# Projecting Assumptions: The Duality Between Sparse Autoencoders and Concept Geometry

**Sai Sumedh R. Hindupur**[1,*], **Ekdeep Singh Lubana**[2,3,*], **Thomas Fel**[4,*], and **Demba Ba**[1,4]

[1]School of Engineering and Applied Science, Harvard University
[2]CBS-NTT Program in Physics of Intelligence, Harvard University
[3]Physics of Artificial Intelligence Group, NTT Research, Inc., Sunnyvale, CA, USA
[4]Kempner Institute, Harvard University

## Abstract

Sparse Autoencoders (SAEs) are widely used to interpret neural networks by identifying meaningful concepts from their representations. However, do SAEs truly uncover all concepts a model relies on, or are they inherently biased toward certain kinds of concepts? We introduce a unified framework that recasts SAEs as solutions to a bilevel optimization problem, revealing a fundamental challenge: each SAE imposes structural assumptions about how concepts are encoded in model representations, which in turn shapes what it can and cannot detect. This means different SAEs are not interchangeable—switching architectures can expose entirely new concepts or obscure existing ones. To systematically probe this effect, we evaluate SAEs across a spectrum of settings: from controlled toy models that isolate key variables, to semi-synthetic experiments on real model activations and finally to large-scale, naturalistic datasets. Across this progression, we examine two fundamental properties that real-world concepts often exhibit: heterogeneity in intrinsic dimensionality (some concepts are inherently low-dimensional, others are not) and nonlinear separability. We show that SAEs fail to recover concepts when these properties are ignored, and we design a new SAE that explicitly incorporates both, enabling the discovery of previously hidden concepts and reinforcing our theoretical insights. Our findings challenge the idea of a universal SAE and underscores the need for architecture-specific choices in model interpretability.

## 1 Introduction

Interpretability has become an important research agenda for assuring, debugging, and controlling neural networks [1–5]. To this end, sparse dictionary learning methods [6–10], especially Sparse Autoencoders (SAEs), have seen a resurgence in literature, since they offer an unsupervised pipeline for simultaneously enumerating all concepts a model may rely on for making its predictions [11–18]. Specifically, an SAE decomposes representations into an overcomplete set of latents that (ideally) correspond to abstract, data-centric concepts which, upon aggregation, explain away the model representations [19, 20]. In other words, an SAE is expected to result in *monosemantic* latents which are more interpretable than the neurons of the original model [21]. For example, SAE latents derived from models in diverse domains have been demonstrated to activate for meaningful concepts such as specific monuments, behaviors, and scripts in language [22, 23]; specific objects, people, and scene properties in vision [15, 24]; and correlate with binding sites and functional motifs in protein autoregressive models [25–27].

To the extent the concepts uncovered using SAEs faithfully represent the concepts used by a model for making its predictions, we can use this information to perform surgical interventions on a model's representations and hence achieve control over its behavior [23, 28, 29]. While this forms a bulk of the motivation around research in SAEs [12, 30, 31], we argue the theoretical foundations that suggest

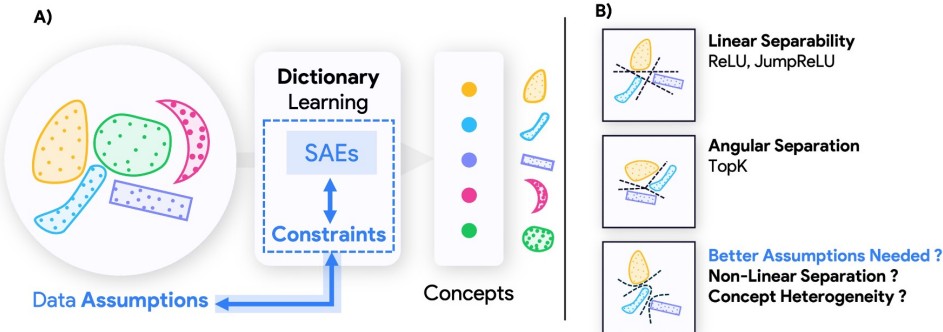

Figure 1: **The Duality Between SAEs Architectures and Their Implicit Data Assumptions. A)** SAEs do not passively extract concepts—they impose constraints that shape what can be detected. Each SAE architecture inherently assumes a specific structure in how features are encoded, leading to a corresponding dual assumption about the data. **B)** Different SAEs rely on different assumptions: some expect features to be linearly separable (ReLU, JumpReLU) or separable by angle while having uniform intrinsic dimensionality (TopK). These assumptions dictate what an SAE can successfully extract—and what it may miss entirely.

SAEs are an optimal tool for achieving this goal are lacking. For example, is it possible that instead of uncovering all concepts a model utilizes in its computation, SAEs are biased towards identifying only a specific, narrower subset of concepts? Furthermore, is it possible that different SAEs, which generally achieve similar fidelity/sparsity, have qualitatively different biases and hence uncover different concepts from model representations? An affirmative answer to these questions may explain recent negative results on SAEs, e.g., algorithmic instability [15, 32] and lack of causality [33, 34]. Motivated by this, we make the following contributions in this work.

- **Duality between Concepts' Organization and the Optimal SAE that Identifies Them.** We formulate SAEs as solutions to a specific bilevel optimization problem, which highlights a fundamental **duality** between concept structure in model representations and an SAE encoder's *receptive fields* (formalized in Def. 3.2). Crucially, this implies any SAE is implicitly biased towards identifying concepts that are organized in a specific manner (Fig. 1).

- **Empirical Validation via Concepts that do not Follow SAEs' Implicitly Assumed Organization.** We evaluate SAEs on concepts with heterogeneous intrinsic dimensionality (i.e., different concepts occupy subspaces of varying dimension) and nonlinear separability through experiments on controlled synthetic setups to real-world model activations, demonstrating that SAEs failing to account for these properties systematically miss the corresponding concepts.

- **A Methodology for Designing Task-Specific SAEs.** Our results suggest no single SAE architecture may be universally optimal, and hence SAEs should be designed by accounting for how concepts are encoded in model representations. To validate this, we introduce SpaDE, a novel SAE that explicitly incorporates heterogeneity and nonlinear separability into its encoder. As we show, SpaDE successfully identifies concepts that other SAEs fail to detect, reinforcing the need for data-aware choices in interpretability.

## 2 Preliminaries

**Notation.** We denote vectors as lowercase bold (e.g., $\boldsymbol{x}$) and matrices as uppercase bold (e.g., $\boldsymbol{X}$). $[n]$ denotes $\{1, \ldots, n\}$ and $\mathcal{B} = \{\boldsymbol{x} \mid \|\boldsymbol{x}\|_2 \leq 1\}$ the unit $\ell_2$-ball in $\mathbb{R}^d$. We assume access to a dataset of $k$ samples, $\boldsymbol{X} = \{\boldsymbol{x}_1, \ldots, \boldsymbol{x}_k\}$, where $\boldsymbol{x} \in \mathbb{R}^d$. For any matrix $\boldsymbol{X}$ or vector $\boldsymbol{x}$, we use $\boldsymbol{X} \geq \boldsymbol{0}$ (resp. $\boldsymbol{x} \geq \boldsymbol{0}$) to indicate element-wise non-negativity.

**Sparse Coding.** Also known as Sparse Dictionary Learning [10, 35], sparse coding assumes a generative model of data as a sparse combination of latents. Specifically, sparse coding involves solving the following optimization problem:

$$\underset{\boldsymbol{z} \geq \boldsymbol{0}, \boldsymbol{D} \in \mathcal{B}}{\arg\min} \quad \sum_{\boldsymbol{x}} \|\boldsymbol{x} - \boldsymbol{D}\boldsymbol{z}\|_2^2 + \lambda \mathcal{R}(\boldsymbol{z}), \tag{1}$$

where $\boldsymbol{z} \in \mathbb{R}^s$ is a sparse latent code, $\boldsymbol{D} \in \mathbb{R}^{d \times s}$ are the dictionary atoms, and $\mathcal{R}(\boldsymbol{z})$ is a sparsity-promoting regularizer, typically $\|\boldsymbol{z}\|_1$. Note that the optimization is performed over *both* the sparse code $\boldsymbol{z}$ (with $\boldsymbol{z} \geq \boldsymbol{0}$) and the dictionary $\boldsymbol{D}$. Further details are included in Appendix A.

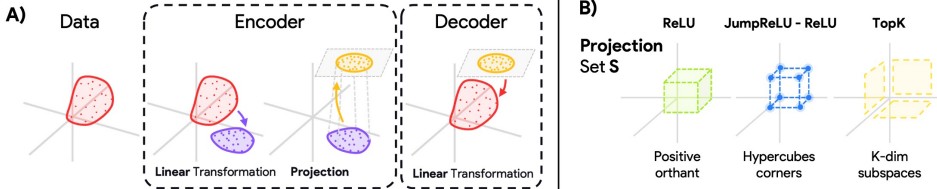

Figure 2: **Projection As The Key Architectural Difference Between SAEs. A)** SAE encoders do more than just linearly transform data—they project it onto an architecture-specific constraint set. This projection fundamentally determines which features an SAE can extract and which it will suppress. **B)** Different SAEs rely on different projection sets $\mathcal{S}$: ReLU projects onto the positive orthant, TopK onto $K-$sparse subspaces, and JumpReLU combines ReLU with a projection onto a hypercube (via a Heaviside step function).

**Sparse Autoencoders.** SAEs [36] approximate sparse dictionary learning by using a single hidden layer to compute the sparse code from data:

$$\text{(i) } \boldsymbol{z} = \boldsymbol{f}(\boldsymbol{x}) = \boldsymbol{g}(\boldsymbol{W}^\top \boldsymbol{x} + \boldsymbol{b}_e), \quad \text{and} \quad \text{(ii) } \hat{\boldsymbol{x}} = \boldsymbol{D}\boldsymbol{z} + \boldsymbol{b}_d, \tag{2}$$

where $\boldsymbol{W}, \boldsymbol{D} \in \mathbb{R}^{d \times s}$ and $\boldsymbol{g} : \mathbb{R}^s \to \mathbb{R}^s$ is the encoder non-linearity. [1] Here, sparsity is enforced on the SAE latent code $\boldsymbol{z}$. SAEs are trained on the sparse dictionary learning loss (Eq. 1), with the sparsity-promoting regularizer $\mathcal{R}$. Different SAEs typically differ in the choice of encoder nonlinearity $\boldsymbol{g}$ and the regularizer $\mathcal{R}$, as discussed in our unified framework next.

# 3 Unified Framework for SAEs

In this section, we develop a framework which captures multiple SAEs used in practice. More specifically, we analyze the following three popular SAE architectures: ReLU SAE [11, 12], TopK SAE [13, 37] and JumpReLU SAE [14, 38]. This framework unravels a duality between how concepts are encoded in model representations and an SAE's architecture. The nonlinearity of the SAEs under study is an orthogonal projection onto some set, where the choice of projection set differentiates SAEs (see Fig. 2). We formalize such nonlinearities as projection nonlinearities, as defined below.

Table 1: **Projection Nonlinearities in SAE Encoders.** Each model can be understood by its nonlinear orthogonal projection $\boldsymbol{g}(\cdot)$ onto a constraint set $\mathcal{S}$ which determines its activation behavior, sparsity structure, and implicit data assumptions.

| Model | $\boldsymbol{g}(\boldsymbol{v})$ |
|---|---|
| ReLU | $\boldsymbol{\Pi}_{\mathcal{S}}\left\{\boldsymbol{v}\right\}, \mathcal{S} = \{\boldsymbol{y} \in \mathbb{R}^s : \boldsymbol{y} \geq 0\}$ |
| TopK | $\boldsymbol{\Pi}_{\mathcal{S}}\left\{\boldsymbol{v}\right\}, \mathcal{S} = \{\boldsymbol{y} \in \mathbb{R}^s : \boldsymbol{y} \geq \boldsymbol{0}, \|\boldsymbol{y}\|_0 \leq k\}$ |
| Heaviside $(H)$ | $\boldsymbol{\Pi}_{\mathcal{S}}\left\{\boldsymbol{v} + \frac{1}{2}\boldsymbol{1}\right\}, \mathcal{S} = \{0, 1\}^s$ |
| JumpReLU | $\text{ReLU}(\boldsymbol{v} - \boldsymbol{\theta}) + \boldsymbol{\theta} \odot H(\boldsymbol{v} - \boldsymbol{\theta})$ |

**Definition 3.1** (Projection Nonlinearity). Let $\boldsymbol{v} \in \mathbb{R}^s$ be a pre-activation vector. A projection nonlinearity $\boldsymbol{\Pi}_{\mathcal{S}}\left\{\cdot\right\} : \mathbb{R}^s \to \mathbb{R}^s$ is defined as:

$$\boldsymbol{\Pi}_{\mathcal{S}}\left\{\boldsymbol{v}\right\} = \arg\min_{\boldsymbol{\pi} \in \mathcal{S}} \|\boldsymbol{\pi} - \boldsymbol{v}\|_2^2, \tag{3}$$

where $\mathcal{S} \subseteq \mathbb{R}^s$ is the constraint set onto which $\boldsymbol{v}$ is orthogonally projected. Popular SAE nonlinearities, e.g., ReLU, JumpReLU, and TopK, are orthogonal projections onto different sets (see Tab. 1).

Generalizing the variational form of projection nonlinearities allows us to formalize SAEs as follows.

**Claim 3.1** (Bilevel optimization of SAEs). *A sparse autoencoder (Eq. 2) with the dictionary learning loss function (Eq. 1) solves the following bi-level optimization problem:*

$$\arg\min_{\boldsymbol{D} \in \mathcal{B}, \boldsymbol{z} \geq \boldsymbol{0}} \sum_{\boldsymbol{x}} \|\boldsymbol{x} - \boldsymbol{D}\boldsymbol{z}\|_2^2 + \lambda \mathcal{R}(\boldsymbol{z})$$

$$s.t. \ \boldsymbol{z} = \boldsymbol{f}(\boldsymbol{x}) \in \arg\min_{\boldsymbol{\pi} \in \mathcal{S}} \boldsymbol{F}(\boldsymbol{\pi}, \boldsymbol{W}, \boldsymbol{x}), \tag{4}$$

*where $\boldsymbol{F}$ is a variational formulation of the SAE encoder $\boldsymbol{f}$. For SAEs, $\boldsymbol{f}(\boldsymbol{x}) = \boldsymbol{g}(\boldsymbol{W}^\top \boldsymbol{x} + \boldsymbol{b}_e)$ (Eq. 2). Note that this inner optimization with the objective $\boldsymbol{F}$ is what differentiates different SAEs.*

---

[1]Encoder bias $\boldsymbol{b}_e$ is not used in the TopK SAE [13].

*Proof.* The outer optimization follows from the dictionary learning loss with sparsity-inducing penalty of the SAE (Eq. 1). The constraint is imposed by the SAE encoder's architecture (Eq. 2). The variational formulation of the encoder as the minimization of some objective $F$ over set $\mathcal{S}$ is a generalization of projection nonlinearities (Eq. 3) for which $F(\pi, W, x) = \|W^\top x + b_e - \pi\|_2^2$. □

This framework implies that each SAE solves a different, constrained (through encoder architecture) optimization version of sparse dictionary learning. *This constraint dictates the quality of the solution obtained*, since it restricts the search space of solutions to dictionary learning, and hence does not have to capture the full sparse coding solution. To further formalize this claim in the next section, we now define receptive fields, a popularly used concept in neuroscience to study the response properties of biological neurons [35]. We use the term *neuron* to define receptive fields in line with the inspiration from neuroscience, but they refer to *neurons of SAEs* (SAE latents) in subsequent analysis.

**Definition 3.2** (Receptive Field). Consider a neuron $k$, which computes a function $f^{(k)} : \mathbb{R}^d \to \mathbb{R}$. The receptive field of this neuron is defined as $\mathcal{F}_k = \{x \in \mathbb{R}^d \mid f^{(k)}(x) > 0\}$.

Intuitively, $\mathcal{F}_k$ represents the region of input space where neuron $k$ is active. *The structure of receptive fields in an SAE is dictated by its encoder's architecture.*

**Duality**: Properties of the SAE encoder will constrain receptive fields' structure for SAE latents. These constraints directly translate to assumptions (often *implicit*, see Sec. 4) about the data structure, since "monosemanticity" [12, 21] requires receptive fields to match structure of concepts in data. Alternatively, if one knows how concepts are organized in the data (model representations), duality can be used to design an appropriate SAE architecture (see Sec. 4.1).

> **Fundamental Limitation of SAEs**
>
> An SAE's encoder enforces *implicit dual assumptions about data*, fundamentally shaping which concepts it can identify and which remain obscure. To build more effective SAEs, these assumptions must *explicitly match the true structure of the data.*

## 4 Implicit SAE Assumptions and Data Properties

In this section, we explicitly state the data assumptions made by ReLU, TopK and JumpReLU SAEs.

**Theorem 4.1** (Implicit Assumptions; Informal). *An SAE makes implicit assumptions about the structure of concepts in data, reflecting it in the receptive fields of its encoder. These assumptions are explicitly stated in Tab. 2 for ReLU, JumpReLU and TopK SAEs (derived in App. D.2).*

The optimality of the above assumptions depends on the "true structure" of concepts in model representations. By "true structure" of concepts, we refer to the ground truth structure in accordance with which concepts are organized in a model's activations. While concept structure is not known in its entirety, we highlight two properties of how (certain) concepts are organized in a model based on recent interpretability literature.

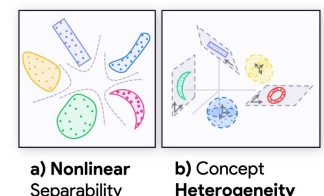

a) Nonlinear Separability    b) Concept Heterogeneity

1. **Nonlinear separability of concepts.** Concepts are not separable by linear decision boundaries. Evidence towards such concepts include features with dependence on magnitude, such as onion features [39]. Even "linear features" [9, 40]), having different magnitudes may fail to be linearly separable (Fig. 3).

Figure 3: **Illustration of Two Reasonable Data Assumptions. A)** Concepts may not be separable using hyperplanes. **B)** Some concepts are inherently low-dimensional, while others span higher-dimensional spaces.

2. **Heterogeneity of concepts.** Different concepts belong to subspaces with different dimensions. Evidence for this property includes unidimensional features representable as concept activation vectors [19], e.g., truth [41], multidimensional features such as days of the week in a 2-D subspace [42], and higher dimensional safety-relevant features [43]. Here, higher dimensional concepts may be compositions of atomic (one-dimensional) concepts (such as safety features composed of "refusal behavior", "hypothetical narrative", and "role-playing" [43]).

Table 2: **Implicit Assumptions of SAEs.** The receptive fields of SAEs implicitly assume concepts are organized with a specific structure in the data, i.e., in model representations.

| Model | Receptive Field | Data Assumption |
|---|---|---|
| ReLU | half-spaces | Linear separability of concepts |
| JumpReLU | half-spaces | Linear separability of concepts |
| TopK | union of hyperpyramids | Angular separability of concepts; same dimensionality per concept |

We characterize the compatibility of different SAEs' implicit assumptions and these properties in Tab. 3. Note that ReLU and JumpReLU can potentially capture heterogeneity since they can show different sparsity levels for each concept, but they require linear separability of concepts due to half-space receptive fields. TopK may be able to handle nonlinear separability to some

Table 3: **Compatibility of SAEs** with nonlinear separability and heterogeneity.

| Model | Nonlinear Sep. | Heterogeneity |
|---|---|---|
| ReLU | ✗ | ✓ |
| JumpReLU | ✗ | ✓ |
| TopK | ✓ | ✗ |

extent (provided concepts are separable by angle), but it cannot adapt to heterogeneous concepts, since it involves a fixed choice of sparsity level for all inputs. BatchTopK [16], a modification of TopK which selects average sparsity level per batch, does not capture concept heterogeneity either, since it still requires choosing the average sparsity level K. To enable evaluation of our claims, we next design an SAE that accommodates the two properties above into its architecture, presented in the following subsection.

## 4.1 SpaDE, or How to Design A Geometry-Driven SAE

We now use the data properties studied above—nonlinear separability and concept heterogeneity—and through the duality, construct one set of sufficient conditions on the SAE to capture both properties, resulting in a novel SAE called SpaDE (Sparsemax Distance Encoder). See App. D.4 for details. We introduce SpaDE as a geometry-driven SAE to validate our claims about the duality between concept geometry and SAE architecture. Hence, SpaDE is expected to capture concepts better than other SAEs when its data assumptions are met.

*Nonlinear separability* can be captured by SAE encoders with a competitive projection nonlinearity (allowing flexible receptive fields, shaped by locations of all weights) and compute Euclidean distances to a set of prototypes instead of linear transforms (to better exploit magnitude). For *concept heterogeneity*, SAEs must demonstrate *adaptive sparsity* in their latent representations, i.e., different concepts must be able to activate different numbers of latents (Fig. 4).

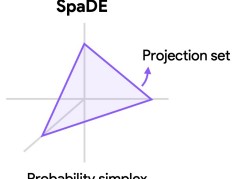

To satisfy the desiderata above, we use a simple first-order equality constraint on the projection set $\mathcal{S}$ (Eq. 4), resulting in the probability simplex $\mathcal{S} = \Delta^s = \{\boldsymbol{x} \in \mathbb{R}^s : \sum_i x_i = 1, \boldsymbol{x} \geq \boldsymbol{0}\}$. The non-negativity is necessary to explain away data as a combination of features with positive contributions. Projection onto the simplex (see Fig. 4) results in the sparsemax nonlinearity ([44]):

$$\text{Sparsemax}(\boldsymbol{v}) = \operatorname*{arg\,min}_{\boldsymbol{\pi} \in \Delta^s} \|\boldsymbol{\pi} - \boldsymbol{v}\|_2^2.$$

The probability simplex $\Delta^s$ admits representations with any (non-zero) level of sparsity, as illustrated in Fig. 4. Combining Sparsemax with euclidean distances then yields SpaDE:

$$\boldsymbol{z} = \boldsymbol{f}(\boldsymbol{x}) = \text{Sparsemax}(-\lambda d(\boldsymbol{x}, \boldsymbol{W})),$$
$$\text{where } d(\boldsymbol{x}, \boldsymbol{W}))_i = \|\boldsymbol{x} - \boldsymbol{W}_i\|_2^2. \tag{5}$$

Figure 4: **SpaDE shows adaptive sparsity by projecting onto the probability simplex**. In this illustrative $3D$ figure, note $\|\boldsymbol{x}\|_0 = 3$ for points on the face, $\|\boldsymbol{x}\|_0 = 2$ for points on edges along subspaces, and $\|\boldsymbol{x}\|_0 = 1$ for corners on coordinate axes.

In the above, $\lambda$ is a scaling parameter (akin to inverse temperature), while $\boldsymbol{W}_i$ is the $i^{th}$ column of the encoder matrix $\boldsymbol{W}$ which behaves as a *prototype* (or landmark) in input space since we compute euclidean distance from input $\boldsymbol{x}$ to $\boldsymbol{W}_i$. App. D.4 and D.2.3 describe the receptive fields of SpaDE in further detail and show how it captures nonlinear separability and concept heterogeneity.

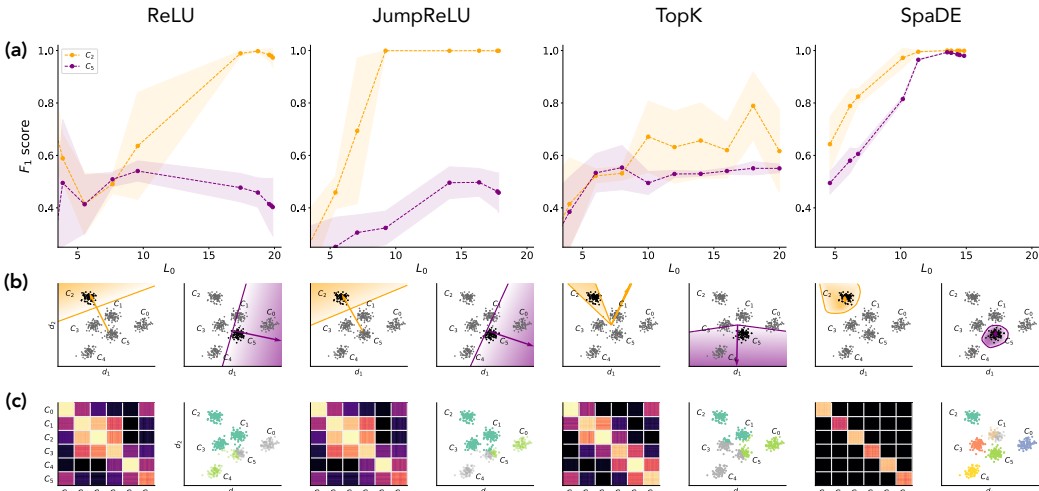

Figure 5: **Effect of Nonlinear Separability on SAEs**. Each column represents a different SAE. **a)** $F_1$ scores of the top 5 most monosemantic latents (highest F1 scores), where shaded region is $\pm$1SD, of each SAE on two concepts—orange (linearly separable) and purple (non-linearly separable). SAEs that assume linear separability struggle to capture the nonlinearly separable concept. **b)** Receptive fields of the most monosemantic latent for each SAE, illustrating how some architectures fail to isolate the nonlinear concept cleanly. Intensity of color indicates strength of SAE latent activation. **(c)** Matrix of pairwise cosine similarities between sparse codes of different datapoints, and data clusters obtained through spectral clustering on this matrix. In the scatter plot, points colored by the same color belong to one spectral cluster, which intuitively indicates that they activate a common set of SAE latents. SpaDE is able to maintain clear concept boundaries and doesn't mix distinct features, while other SAEs group subsets of different features into the same spectral cluster (same color).

We also note that the outer optimization for SpaDE is K-Deep Simplex (KDS, [45]), a modified dictionary learning technique which incorporates locality into sparse representations. The regularizer from KDS is a distance-weighted $\ell_1$ regularizer $\mathcal{R}(\boldsymbol{z}) = \sum_i z_i \|\boldsymbol{x} - \boldsymbol{W}_i\|_2^2$, which encourages prototypes to move closer to data when they are active, increasing sparsity of representation [2]. The inner optimization for SpaDE is a one-sided sparsity-regularized optimal transport (see App. D.4).

Our claim about the duality between SAE architectures and data assumptions about concepts also applies to SpaDE. Beyond nonlinear separability and concept heterogeneity, SpaDE implicitly assumes that Euclidean distances are useful in concept space—concepts are distance-separated—and distances can be used to disentangle concepts.

# 5 Results: Empirical Validation of SAE behavior

We perform a suite of experiments which involve training ReLU, JumpReLU, TopK and SpaDE SAEs on synthetic Gaussian clusters, semi-synthetic formal-language model activations and natural vision model activations. Our synthetic experiments aim to validate our claims about implicit assumptions in SAEs. Experiments on more naturalistic data seek to demonstrate our claims extend to realistic data settings. Further analysis is deferred to App. E. The code to replicate synthetic experiments is available at: `https://github.com/Sai-Sumedh/SaeConceptDuality-SpaDE`, formal language experiments is at: `https://github.com/EkdeepSLubana/spadeFormalGrammars`, and vision experiments is at: `https://github.com/KempnerInstitute/Overcomplete`.

## 5.1 Separability Experiment

**Dataset and Experiment**: We construct a 2-dimensional dataset with Gaussian clusters (abstraction of concepts) of different magnitudes in order to demonstrate nonlinear separability of concepts in a

---

[2]This regularizer encourages dictionary atoms to "stick" to the data, addressing the recently raised concern [15, 32] that directions learned by SAEs may be out-of-distribution (OOD), contributing to their instability.

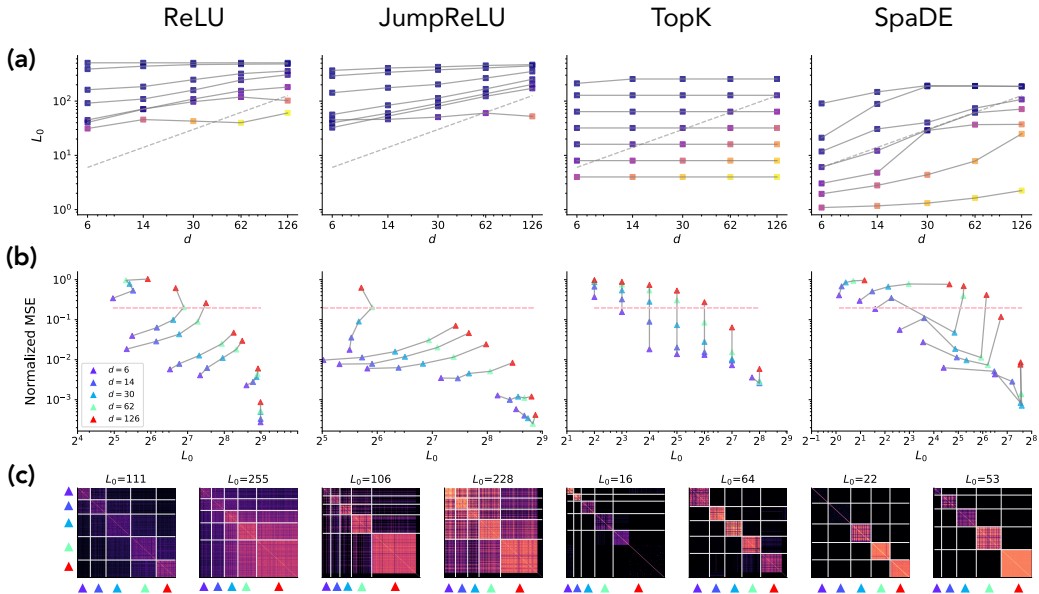

Figure 6: **Effect of Concept Heterogeneity on SAEs**. **a)** Per-concept sparsity as a function of intrinsic dimension. Colors indicate per-concept MSE—higher errors (red/yellow) show when an SAE fails to capture a concept effectively. Each solid line indicates one model with a specific choice of hyperparameters. **b)** Normalized MSE vs. per-concept sparsity. A well-performing SAE should maintain low error across all concepts. TopK SAE only achieves good reconstruction (below the dashed 20% error threshold) when sparsity (fixed for a given model) exceeds intrinsic dimensionality, highlighting its lack of flexibility. **c)** Cosine similarity between pairs of SAE latents across all concepts (showing co-occurrence), visualized for two sparsity levels.

simple setting which facilitates visualization. Here, each cluster is defined as its own concept, and we expect SAEs to learn latents responding to individual clusters. The concepts with smaller norm are not linearly separable, while those with larger norm are linearly separable. We train all SAEs on this dataset for a range of sparsity levels. Following our arguments about implicit assumptions in SAEs, we hypothesize that ReLU and JumpReLU will be unable to capture the nonlinearly separable concepts with monosemantic latents (measured using F1 scores; see Eq. 9).

**Observations**: Fig. 5 shows how different SAEs fare on this experiment. ReLU and JumpReLU achieve an F1 score of 1 for the separable concept (orange), while their F1 scores are much lower and bounded above (by 0.5) for the nonlinearly separable concept (purple). The receptive fields of ReLU, JumpReLU in Row (b) clearly overlap with other concepts in the nonlinearly separable case. TopK performs somewhat poorly on both concepts, showing comparable F1 scores in both cases. SpaDE shows a top F1 score of 1.0 for both concepts (perfect precision and recall), with its receptive fields capturing concept structure even for nonlinearly separable concepts. While ReLU and JumpReLU show significant cross-concept correlations (between concepts $C_1, C_2, C_3$, Row (c)) and TopK does marginally better with smaller correlations, SpaDE shows clear delineation of different concepts with clearly separated concepts (no cross correlations, spectral clustering identifies concepts). Note SpaDE may overspecialize and lead to further subclusters, as seen by two colors within concept 1 in row (c).

## 5.2   Heterogeneity Experiment

**Dataset and Experiment**: We generate Gaussian clusters (again an abstraction for concepts) in a 128-dimensional space. The five concepts are heterogeneous—they belong to subspaces with different intrinsic dimensions (6, 14, 30, 62, 126), but are designed to have isotropic structure within each cluster, and similar total variances across clusters. We trained ReLU, JumpReLU, TopK SAEs and SpaDE on this data with varying sparsity levels. We hypothesize that TopK will not be able to adapt its representations to the intrinsic dimension of each cluster.

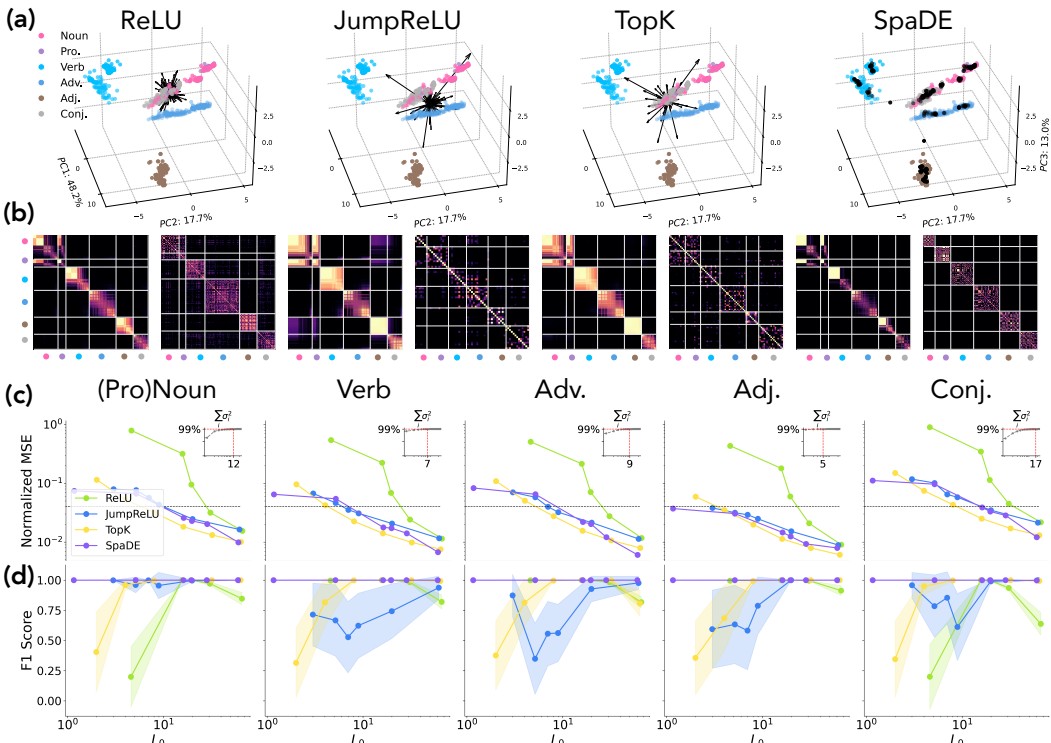

Figure 7: **Investigating SAE properties on GPT for formal languages**. **(a)** 3D PCA of model activations and SAE encoder weights, where datapoints are colored by part-of-speech (PoS). Encoder weights are indicated by points for SpaDE and arrows for the other SAEs. **(b)** Matrix of cosine similarities between pairs of data and pairs of latents (in order) for each SAE. White lines separate different PoS. **(c)** MSE normalized by PoS variance as a function of sparsity, for each PoS. *Inset:* cumulative sum of variance (eigenvalues of data correlations) of each PoS, where the effective dimension (variance $> 99\%$) of each PoS is shown. **(d)** Top-20 $F_1$-scores for different PoS from each SAE's latents (a measure of monosemanticity).

**Observations**: Fig. 6 shows the results of all SAEs on this experiment. In Row (a), TopK shows the same level of sparsity per concept for all concepts, along with worse reconstruction error for higher dimensional concepts. In contrast, other SAEs—ReLU, JumpReLU and SpaDE—show adaptive sparsity to different extents by adjusting their representations to the intrinsic dimension of each concept. SpaDE can capture the intrinsic dimension nearly perfectly (along the dashed $y = x$ curve) for a specific choice of hyperparameters.

Note that a naïve estimator which predicts the mean of each concept will achieve a normalized MSE of 1. For TopK, normalized MSE (Row (b)) goes below $20\%$ (i.e., explains $80\%$ of the variance) for each concept only when $k$ exceeds the dimension of that concept. For example, $d = 6$ goes below the dashed line only after $k = 8$, similarly for other concepts. Other SAEs are able to stay below the $20\%$ threshold for nearly all concepts across hyperparameters.

In Row (c), each latent is assigned a concept which it activates maximally for. Note that different concepts use different numbers of latents in ReLU, JumpReLU, and SpaDE. However, there are correlations across concepts in ReLU and JumpReLU (for the dense case), indicating co-occurrence of latents across concepts, which is reduced in the sparse case. Correlations are absent in SpaDE under both cases. TopK uses similar number of latents across concepts, inline with its lack of adaptivity.

## 5.3 Formal Languages

**Dataset and Experiment**: Building on recent work using formal languages for making predictive claims about language models [46–48], we use this setting as a semi-synthetic setup for corroborating our claims. Specifically, we analyze the English PCFG (Probabilistic Context-Free Grammars, formal

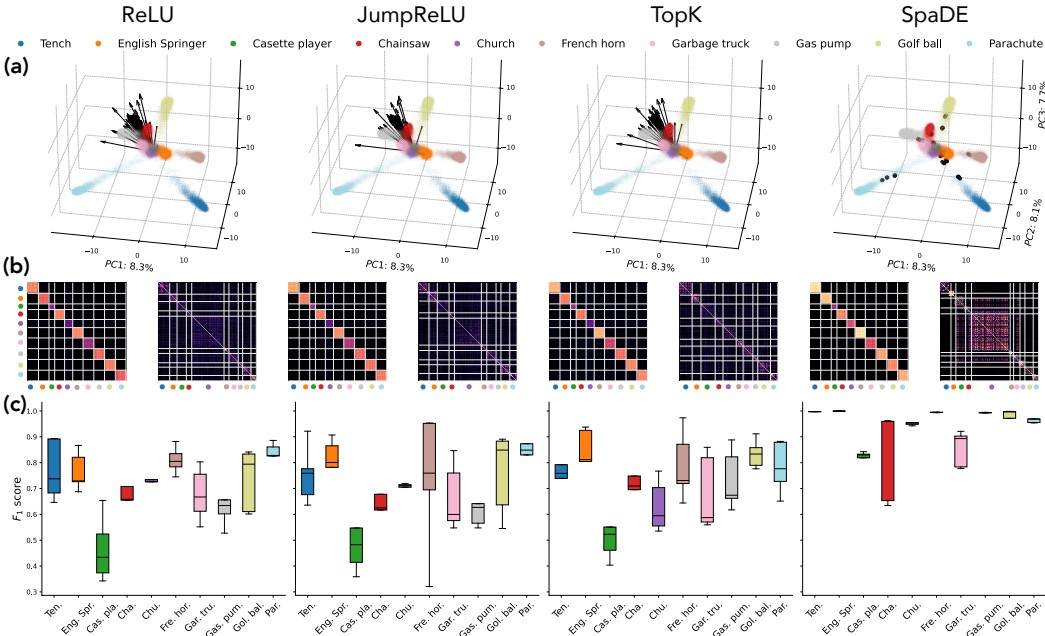

Figure 8: **SAE properties on DINOv2 activations**. **(a)** 3-D PCA of model activations colored by class, and SAE encoder weights (points for SpaDE, arrows for other SAEs). **(b)** Cosine similarities of sparse codes of pairs of data and pairs of SAE latents (in order) for each SAE. White lines separate classes. **(c)** $F_1$ scores of top-5 most monosemantic latents for each SAE across classes (color-coded)

models of language often used to study its syntactic properties, see App. C.3) with subject-verb-object sentence order proposed in [34]. We train 2-layer Transformers [49] from scratch on strings of maximum length 128 tokens from the formal grammar above. SAEs are then trained on activations retrieved from the middle residual stream of the model.

**Observations**: Results are shown in Fig. 7. Different parts of speech (PoS), the core concepts of the grammar, form clusters in a 3D PCA of their representations (see row (a)). SpaDE learns to tile the PoS clusters well. While all SAEs do a good job at making their latents uncorrelated across PoS (first column per SAE, row (b)), there are co-occurring latents across PoS in all SAEs except SpaDE (second column per SAE, row (b)). PoS seem to have different intrinsic dimensions (number of dimensions to capture 99% of total variance in data, inset in row (c)), which leads to TopK requiring different values of K to explain the data (crosses 5% normalized MSE with differing values of k, row (c)). PoS also appear to have differing levels of linear separability, as ReLU and JumpReLU show lower F1 scores which peak at different levels of sparsity for each concept (row (d)), while SpaDE shows a perfect F1 score of 1 in its most monosemantic latents.

### 5.4 Vision

**Dataset and Experiment:** We use *Imagenette*, a 10-class subset of ImageNet [50], containing 1.5k images per class. Representations are extracted from the *DINOv2-base* (with registers), yielding 261 tokens per image. Over the course of 50 training epochs, this yields approximately 200 million tokens. SAEs are trained on all available tokens, including spatial, CLS, and registers tokens, for 50 epochs with 200 latent dimensions.

**Observations**: Results are shown in Fig. 8. SpaDE again tiles the class structure well in the 3-D PCA (row (a)). Similarities between sparse codes of data (first column of each SAE in row (b)) show that all SAEs are able to decorrelate different classes in their latent representations. Latent co-occurrence (second column of each SAE in row (b)) is widespread in ReLU, JumpReLU and TopK SAEs, but it seems to be specific to certain pairs of latents in SpaDE. $F_1$ scores (row (c)) show that SpaDE has the most monosemantic latents across all classes. The varying $F_1$ scores for ReLU and JumpReLU across classes indicate different levels of linear separability across classes. Importantly, we find SpaDE identifies interpretable concepts such as foreground/background, different parts of objects in

an image (hands, face, fins of fish, windows/ stairs in church images, eyes, ears, snout of dogs, etc), which are visualized using feature attribution maps in App. E.4.

# 6 Discussion and Limitations

Our findings reveal critical insights into the limitations and strengths of different sparse autoencoder (SAE) architectures for concept discovery. We observed that ReLU and JumpReLU SAEs fail to capture nonlinear separability ( low F1 scores, latent co-occurrence across concepts), while Top-K SAE struggles to capture concept heterogeneity (high MSE when concept dimension exceeds choice of K). A common issue across these architectures is the co-occurrence of latents across multiple concepts, indicating a lack of concept specialization. In contrast, SpaDE achieves the highest F1 scores for its most monosemantic latents, exhibits low latent co-occurrence across concepts and enforces adaptive sparsity, making it an effective choice for structured concept representations. Our observations about the limitations of ReLU, JumpReLU and TopK SAEs highlight that the failure modes of different SAEs stem from a mismatch between their inductive biases and the true structure of the data. Specifically, ReLU and JumpReLU assume linear separability of concepts, which does not always hold, even for concepts that correspond to specific directions in the latent space. On the other hand, our results suggest that incorporating data geometry into SAE design significantly improves concept specialization of SAE latents, allowing it to learn latents that are better aligned with the data.

Overall, our results emphasize that there may not be a single best SAE architecture for all contexts unless the architecture explicitly integrates a sufficient set of data properties relevant to the specific problem. This suggests a shift in focus from using generic SAEs to tailoring their design based on prior knowledge about the underlying data geometry.

Our analysis of receptive fields of SAE encoders and their relation with concept geometry to study monosemanticity properties is quite general, and can also be used to study other kinds of interpreter models such as transcoders [51, 52].

**Limitations:** While SpaDE demonstrates promising improvements over ReLU, JumpReLU and TopK SAEs in synthetic, semi-synthetic and realistic data, we do not claim it to be the optimal SAE for all scenarios. Instead, we present it as a concrete example of how incorporating reasonable data properties (nonlinear separability and concept heterogeneity) can improve interpretability. Thus, several limitations remain, as follows.

- Data properties beyond those considered here may be crucial for improved SAE performance. Future work may explore additional geometric structure of concepts in neural networks to design better SAEs.
- SpaDE implicitly assumes concepts are separated by Euclidean distance, which may still result in latent co-occurrence if concepts do not satisfy this assumption.
- Overly specialized latents may emerge in SpaDE if the sparsity level is too aggressive, potentially leading to latents that capture special cases rather than generalizable concepts.
- We have focused our attention on mutually exclusive concepts, where the presence of one concept implies the absence of others. While our arguments about SAE assumptions hold even when concepts overlap, the expected co-occurrence structure may differ in such cases. For co-occurring concepts, our receptive field analysis can be applied to the presence/absence of each concept.

Overall, we note our work is not a proposal for the best SAE, but a guiding framework for improving design of SAEs that yield useful interpretations demonstrating how better integration of data geometry can enhance model interpretability. The interpretability community may need to prioritize a deeper understanding of latent space geometry, and translate novel insights into SAE design, leading to models with more faithful and structured representations of concepts.

## Acknowledgments

This work has been made possible in part by a gift from the Chan Zuckerberg Initiative Foundation to establish the Kempner Institute for the Study of Natural and Artificial Intelligence at Harvard University. ESL thanks Hidenori Tanaka and the Physics of Intelligence group at CBS-NTT program Harvard for useful conversations and access to compute resources. All authors thank the CRISP group at Harvard SEAS for insightful conversations.

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

# A  Dictionary Learning

Sparse coding [10] (alternatively known in this work as sparse dictionary learning, or just dictionary learning) was initially proposed to replicate the observed properties ("spatially localized, oriented, bandpass receptive fields") of biological neurons in the mammalian visual cortex. It aims to invert a linear generative model with a sparsity prior on the latents:

$$\boldsymbol{x} = \boldsymbol{D}^* \boldsymbol{z}^* + \eta$$

where $\boldsymbol{x} \in \mathbb{R}^n$ is the data, $\boldsymbol{D}^* \in \mathbb{R}^{n \times s}$ is the set of $s$ dictionary atoms, $\boldsymbol{z}^* \in \mathbb{R}^s_+$ is the sparse code, and $\eta$ is additive white Gaussian noise. Given data $\{\boldsymbol{x}^{(1)}, \ldots, \boldsymbol{x}^{(P)}\}$, sparse coding performs maximum aposteriori (MAP) estimation for the dictionary $\boldsymbol{D}^*$ and representations $\boldsymbol{z}^*$ under suitably defined prior and likelihood functions [53] by solving the following optimization problem (repeated from Eq. 1):

$$\underset{\boldsymbol{D} \in \mathcal{B}, \boldsymbol{z}^{(\cdot)} \geq 0}{\arg\min} \sum_k \|\boldsymbol{x}^{(k)} - \boldsymbol{D}\boldsymbol{z}^{(k)}\|_2^2 + \lambda \mathcal{R}(\boldsymbol{z}^{(k)}) \tag{6}$$

where $\mathcal{R}(\cdot)$ is a sparsity-promoting regularizer. The set $\mathcal{B} \subseteq \mathbb{R}^{n \times s}$ includes restriction to unit norm (typical). Generally, the L1 penalty is used as the regularizer term, i.e., $\mathcal{R}(\boldsymbol{z}^{(k)}) = \|\boldsymbol{z}^{(k)}\|_1$, since using the L0 penalty makes the problem NP-hard [54]. When the number of dictionary atoms is less than or equal to the dimension of input space, $s \leq n$, this is an undercomplete problem, and the sparse code can be readily obtained using the pseudo-inverse of the dictionary matrix $D$ (provided the dictionary atoms are linearly independent), leading to the solution $\boldsymbol{z} = (\boldsymbol{D}^T \boldsymbol{D})^{-1} \boldsymbol{D}^T \boldsymbol{x}$. Note that in this (undercomplete) case, the sparse code is a linear transformation of the input. The more interesting setting involves using an overcomplete dictionary ($s > n$), and was initially studied in [35]. Obtaining the sparse code $\boldsymbol{z}$ from input data $\boldsymbol{x}$ is nontrivial in this case.

In this case, sparse coding results in a sparse representation of the data and a dictionary which behaves as a data-adaptive basis. Correspondingly, sparse codes have been shown to capture interesting concepts in data [55, 56], e.g., responding to wavelet-like regions when trained on natural images [57]. In this (overcomplete) setting, a popular approach is using iterative shrinkage and thresholding algorithms (ISTA) [58] and their variants such as FISTA (Fast ISTA) [59]. Modern approaches to this problem use ISTA to design deep residual networks with shared weights and train the network on the sparse coding objective, in a technique called Learned ISTA (LISTA) [60]. Algorithm unrolling [61] is a generalization of this technique and involves designing *interpretable* neural networks using iterative algorithms where each layer of the network reflects an iteration of the algorithm. These networks are interpretable since the weights correspond to an underlying process which was used to design the iterative algorithm. Unrolling has widespread applications in signal processing, and is extensively reviewed in [61].

We also note that sparse coding has been used with algorithm unrolling as a model-based interpretable deep learning technique for a wide range of applications, including image super-resolution [62], graph signal denoising [63], mechanical fault diagnosis [64], deconvolving neural activity of dopamine neurons in mice [65]. Therefore, assuming a linear generative model of data (Eq. 1) where the dictionary atoms are physically relevant in some application, sparse coding using an unrolled network learns the underlying interpretable dictionary atoms.

# B  Related Work

SAEs are a specific instantiation of the broader agenda of dictionary learning tools for concept-level explainability [7–9, 19, 20, 66]. A number of SAE architectures have been proposed recently, including ReLU SAE [12], TopK SAE [13, 37], gated SAE [67], JumpReLU SAE [14], Batch TopK SAE ([16]), ProLU SAE ([68]), and so on. While promising results have been discovered, e.g., latents that respond to concepts of refusal, gender, text script [12, 22, 23], foreground vs. background concepts [17], and concepts of protein structures [25–27], a series of negative results have started to emerge on the limitations of SAEs. For example, [33, 69] show a mere prompting baseline can outperform model control compared to SAE or probing based feature ablation baseline. Similar results were observed by [34] in a narrower formal language setting. Meanwhile, criticizing the underlying linear representation hypothesis that has informed design of earlier SAE architectures (specifically, the vanilla ReLU SAEs), [42, 70] has shown that SAE features can be multidimensional

and nonlinear. Importantly, recent results from [15, 32, 71] have shown that two SAEs trained on the exact same data, just with a different seed, can yield very different concepts and hence very different interpretations. These results are related to the lack of canonical nature in SAE latents [72] This behavior, often called algorithmic instability, makes reliability of SAEs challenging for any practical purposes. More broadly, given the hefty research investment going into the topic, we believe it is warranted that a more formal and theoretical account help solidify the limitations and challenges SAEs (or at least the current paradigm thereof) faces. This can help steer the research in a direction that yields meaningful improvement in SAEs, e.g., in their practical utility. This motivation underscores our work. For a related effort on this front, we highlight the work by [73], who contextualize SAEs from a minimum-description length perspective and enable an intuitively solid account of how features may split to overly specialized concepts (e.g., tokens).

**Disentangled Representation Learning.** As mentioned in Sec. 3, results similar to ours have been reported in the field of disentangled representation learning, wherein one aims to invert a data-generating process to identify the factors of variants (i.e., latent variables) that underlie it. To this end, autoencoders were used as a popular tool, since they offer a method that can (ideally) simultaneously invert the generative process and identify the underlying latents [74]. However, [75] showed that in fact this problem is rather challenging: unless one designs an autoencoder architecture that bakes-in assumptions about the generative process, i.e., the precise function mapping itself, there are no guarantees the retrieved latents will correspond to the ground-truth ones. This result led to design of several methods focused on exploiting "weak supervision", i.e., extra information available from data-pairs such as multiple views of an image or temporally consistent video frames, to circumvent the theoretical challenges of disentanglement [76–78]. Our contributions are similar in nature to these results on disentanglement, but we (i) specifically focus on the context of SAEs and (ii) provide a more concrete proof that establishes precisely what the inductive biases of popular SAEs are, i.e., what concepts the SAEs are biased towards uncovering. Having established these results, we now believe the next step that the disentanglement community took, i.e., use of weak supervision, would make sense for the SAEs community as well. This can involve exploiting temporal correlations between tokens in a sentence, or the fact that representations across layers do not change much, as in Crosscoders and Transcoders [51, 52, 79].

## C    Experimental Setup

The synthetic experiments (separability, heterogeneity) and vision experiments were run on NVIDIA A100 40GB GPUs, while the formal language experiments were run on NVIDIA RTX A6000 48GB GPUs.

### C.1    Separability experiment

We construct a synthetic dataset consisting of six isotropic Gaussian clusters in a two-dimensional (2D) space. The cluster centers are arranged such that adjacent clusters are separated by an angular difference of $2\pi/6$, with alternate clusters having norms of 1 and 3. Each cluster is sampled from a multivariate normal distribution with a variance of $2^{-5.5}$. The dataset consists of 1 million data points per concept, yielding a total of 6 million samples. Of these, we use 70% (700,000 points) for training.

Our experiments evaluate four sparse autoencoder (SAE) architectures: ReLU SAE, JumpReLU SAE, TopK SAE, and SpaDE. The first three architectures are implemented following their original formulations (in [12],[14],[13]), with the decoder activations normalized in the forward pass. The SpaDE model follows the same single hidden-layer autoencoder structure but differs in that it utilizes Euclidean distance computations and a SparseMax activation function for the encoder. Across all models, the hidden-layer width is set to 128, and a pre-encoder bias is used in all cases except for SpaDE.

For training, the (inverse) temperature parameter $\lambda$ in SpaDE is initialized to $1/(2 \times \text{input dimension})$ and parameterized using the Softplus function to ensure non-negativity. This parameter trained along with the encoder and decoder weights, to allow the model to *learn* its desired sparsity level. Note that large values of $\lambda$ lead to greater sparsity since Sparsemax is scale-sensitive. In JumpReLU, the threshold is initialized at $10^{-3}$ across all latent dimensions, with a bandwidth of $10^{-3}$ for the straight-through estimator (STE), as it is proposed in [14]. All models are trained using the

Adam optimizer with a learning rate of $10^{-2}$, which follows a cosine decay schedule from $10^{-2}$ to $10^{-4}$. The momentum parameter is set to 0.9, and we use a batch size of 512. Training runs for approximately 8000 iterations, and gradient clipping is applied (gradient norms are clipped at 1) to stabilize optimization.

Regularization parameters are selected such that sparsity levels remain comparable across models. Specifically, the regularization coefficient $\gamma$ is chosen in the range $10^{-6}$ to 1 for ReLU and JumpReLU SAEs, between 4 and 64 (powers of 2) for TopK SAE, and in the range $10^{-6}$ to 1 for SpaDE. Each model applies a different regularization strategy: ReLU SAE uses $L_1$ regularization, JumpReLU SAE applies $L_0$ regularization with a straight-through estimator (STE) as in [14], TopK SAE does not use explicit regularization but incorporates an auxiliary loss term as in [13], with $K_{aux} = k$ (same as the choice of sparsity level $k$ in TopK) with $\gamma_{\text{aux}} = 1$ (the scaling for the auxiliary loss term), and SpaDE employs a distance-weighted $L_1$ regularization, which comes from [45].

All networks are initialized such that the decoder weights are initially set as the transpose of the encoder weights, though they are allowed to update freely during training. Model weights are sampled from a normal distribution $\mathcal{N}(0, 1)$. To maintain consistency in scale between inputs and latent activations, a scaling factor $\lambda$ is applied to all latent units, given by $\lambda \approx 1/2 \times$ input dimension (note that this is not trainable for ReLU, JumpReLU and TopK SAEs). Across all architectures, we use the Mean Squared Error (MSE) loss function, with the regularizers and regularizer scaling constants as described above.

For evaluation, we analyze a subset of 1000 data points per concept. The primary metric for comparison is the F1-score, which is computed based on precision and recall. Precision is defined as:

$$\text{Precision} = \frac{\text{True Positives}}{\text{True Positives} + \text{False Positives}}, \tag{7}$$

while recall is given by:

$$\text{Recall} = \frac{\text{True Positives}}{\text{True Positives} + \text{False Negatives}}. \tag{8}$$

Using these definitions, the F1-score is computed as:

$$F1 = \frac{2 \times \text{Precision} \times \text{Recall}}{\text{Precision} + \text{Recall}}. \tag{9}$$

In our setup, precision and recall are computed by thresholding latent activations at $10^{-6}$. Additionally, we analyze the receptive fields by creating a 2D meshgrid, passing all points through the model, and extracting their SAE latent representations. Cosine similarities between pairs of data points are also computed by obtaining their latent representations, calculating the pairwise cosine similarity, and organizing the results by class.

To further examine latent space structure, we compute the stable rank of the representation matrix. Stable rank for the similarity matrix is computed as the sum of singular values divided by the largest singular value (alternatively called the intrinsic dimension of this matrix):

$$\text{Stable Rank} = \frac{\sum \sigma_i}{\sigma_{\max}}. \tag{10}$$

Finally, spectral clustering is performed on the similarity matrix derived from latent representations. The number of clusters is determined by the stable rank of this similarity matrix (rounded up), providing insights into the correlations between SAE latent representations.

## C.2 Heterogeneity experiment

We construct a synthetic dataset consisting of five isotropic Gaussian clusters in a 128-dimensional space. The intrinsic dimensionality of each cluster follows the sequence $2^q - 2$ for different values of $q \in \{3, 4, 5, 6, 7\}$, resulting in clusters with intrinsic dimensions of $6, 14, 30, 62, 126$, respectively.

The lower-dimensional clusters belong to subspaces that form strict subsets of the subspaces of higher-dimensional ones, meaning that the first six dimensions are fully contained in the next 14, which are further contained in the next 30, and so on up to 126 dimensions. Cluster centers are sampled uniformly at random from the range $[0, \frac{1}{21}]$ along each dimension. The variance of each concept is chosen to be inversely proportional to its intrinsic dimension to ensure that the total variance per concept remains constant across all concepts. The dataset contains 6.4 million data points per concept, yielding a total of 32 million samples, of which 70% (approximately 22 million points) are used for training.

Our models follow four different sparse autoencoder (SAE) architectures: ReLU SAE, JumpReLU SAE, TopK SAE, and SpaDE. The first three are implemented according to their original formulations in [12], [14], and [13], with the decoder activations normalized in the forward pass. The SpaDE model follows the same single hidden-layer autoencoder structure but differs in that it utilizes Euclidean distance computations and a SparseMax activation function for the encoder. Across all models, the SAE hidden-layer width is set to 512. A pre-encoder bias is applied in all cases except for SpaDE. Additionally, for the TopK SAE, a ReLU activation is applied before selecting the top $k$ latent dimensions.

For training, the temperature parameter $\lambda$ in SpaDE is initialized at $1/(2 \times$ input dimension$)$ and parameterized using the Softplus function to ensure non-negativity. This parameter trained along with the encoder and decoder weights, to allow the model to *learn* its desired sparsity level. In JumpReLU, the threshold is initialized at $10^{-3}$ across all latent dimensions, with a bandwidth of $10^{-3}$ for the straight-through estimator (STE). All models are trained using the Adam optimizer with a learning rate of $10^{-2}$, which follows a cosine decay schedule from $10^{-2}$ to $10^{-4}$. The momentum parameter is set to 0.9, and we use a batch size of 2048. Training runs for approximately 10,000 iterations, and gradient clipping (restricting gradient norms to be less than 1) is applied to stabilize optimization.

Regularization parameters are selected such that sparsity levels remain comparable across models. Specifically, the regularization coefficient $\gamma$ is chosen in the range $10^{-3}$ to 5.0 for ReLU SAE, $10^{-3}$ to 1 for JumpReLU SAE, from 4 to 256 (powers of 2) for TopK SAE, and from $10^{-3}$ to 10 for SpaDE. Each model applies a different regularization strategy: ReLU SAE uses $L_1$ regularization, JumpReLU SAE applies $L_0$ regularization with a straight-through estimator (STE) following from [14], TopK SAE does not use explicit regularization but incorporates an auxiliary loss term with $\gamma_{\text{aux}} = 1$ (scaling for the auxiliary term in the loss) and $K_{aux} = k$ (same as sparsity level), and SpaDE employs a distance-weighted $L_1$ regularization.

All networks are initialized such that the decoder weights are initially set as the transpose of the encoder weights, though they are allowed to update freely during training. Model weights are sampled from a normal distribution $\mathcal{N}(0, 1)$. To maintain consistency in scale between inputs and latent activations, a scaling factor $\lambda$ is applied to all latent units, given by $\lambda \approx 1/2 \times$ input dimension. Across all architectures, we use the Mean Squared Error (MSE) loss function.

For evaluation, we analyze a subset of 1000 data points per concept. We report the *normalized MSE*, defined as the ratio of the standard MSE to the variance of the corresponding concept:

$$\text{Normalized MSE} = \frac{\text{MSE}}{\text{Variance of Concept}}. \tag{11}$$

We also compute *sparsity* ($L_0$) per concept, measured as the average number of active latents per data point, averaged over each concept.

To analyze latent representations, we examine cosine similarities in two contexts: (i) between pairs of SAE latent representations for different input data points (per-input co-occurrence) and (ii) between pairs of latents aggregated over all data points (global co-occurrence). For the latter, each latent is assigned a *concept label* based on the concept for which it is most frequently activated on average. This assignment provides insight into how latents specialize across different underlying structures in the dataset.

### C.3 Formal Languages experiment

**Data.** The formal language setup analyzed in the main paper (Sec. 5.3) involves training a 2-layer nanoGPT model on strings from an English-like PCFG (Probabilistic Context-Free Grammars).

Broadly, a PCFG is defined via a 5-tuple $G = (\text{NT}, \text{T}, \text{R}, \text{S}, \text{P})$, where NT is a finite set of non-terminal symbols; T is a finite set of terminal symbols, disjoint from NT; R is a finite set of production rules, each of the form $A \to \alpha\beta$, where $A \in \text{NT}$ and $\alpha, \beta \in (\text{NT} \cup \text{T})$; $S \in \text{NT}$ is the start symbol; and P is a function $\text{P} : \text{R} \to [0, 1]$, such that for each $A \in \text{NT}$, $\sum_{\alpha:A \to \alpha \in \text{R}} \text{P}(A \to \alpha\beta) = 1$. To *generate* a sentence from the grammar, the following process is used.

1. Start with a string consisting of the start symbol $S$.
2. While the string contains non-terminal symbols, randomly select a non-terminal $A$ from the string. Choose a production rule $A \to \alpha\beta$ from R according to the probability distribution $\text{P}(A \to \alpha)$.
3. Replace the chosen non-terminal $A$ in the string with $\alpha$, the right-hand side of the production rule.
4. Repeat the production rule selection and expansion steps until the string contains only terminal symbols (i.e., no non-terminals remain).
5. The resulting string, consisting entirely of terminal symbols, is a sentence sampled from the grammar.

We follow the same rules of the grammar considered in [34]. The strings are tokenized via one-hot encoding via a manually defined tokenizer.

**Model training.** Models are trained from scratch on strings sampled from the grammar above. Strings are padded to length 128 (if not already that length), and a batch-size of 128 ($\sim$10K tokens per batch) is used for training. Training uses Adam optimizer with a cosine learning-rate schedule starting at $10^{-3}$ and ending at $10^{-4}$ after 70K iterations, alongside a weight decay of $10^{-4}$. The nanoGPT models used in this work have a width of 128 units, with an MLP expansion factor of 2 and also 2 attention heads per attention layer.

**SAE training.** All SAEs trained in the formal language setup involve an expansion factor of $2\times$, i.e., 256 latents for a residual stream of 128 dimensions. Training involves a constant learning rate of $10^{-3}$ and lasts for 10K iterations ($\sim$1M tokens). We sweep regularization strength for SAEs' training, yielding SAEs with different sparsity levels. While we fix the regularization strength for SpaDE based on best values identified from the synthetic, Gaussian cluster datasets, for other SAEs (ReLU, JumpReLU, and TopK) we report the best possible results from our sweep by looking at the top-10 per-concept F1 scores; i.e., reported results are a best-case estimate of results achievable by training of these SAEs, and in practice performance can be expected to be poorer than what we analyze. Cross-task transfer for SpaDE's hyperparameters is intriguing in this regard, since we found other SAEs' hyperparameters to not transfer.

### C.4 Vision experiment

**Data.** We use an off-the-shelf, large-scale pretrained model for our analysis in these experiments, specifically *DINOv2-base* (with registers). For simplicity, we focus on a 10-class subset of ImageNet, called *Imagenette*, containing 1.5k images per class. Representations are extracted from the model for images of these classes, yielding 261 tokens per image.

**SAE training.** SAEs are trained on all available tokens, including spatial, CLS, and registers tokens, for 50 epochs with 200 latent dimensions. With 261 tokens per image, this amounts to $\sim$200M tokens for training SAEs over the course of 50 training epochs. For each SAE, the best reconstruction is selected based on a sparsity-controlled learning rate sweep. This resulted in an optimal learning rate of $5 \times 10^{-4}$ for TopK, ReLU, and SpaDE, while JumpReLU performed best with $10^{-4}$ (using Adam optimizer). Additionally, we note our JumpReLU implementation employs a Silverman kernel with a bandwidth of $10^{-2}$, which we found to work best for our setting.

## D Further Theory Results

### D.1 Projections and Nonlinearities

The nonlinearity of popular SAEs is commonly an orthogonal projection onto some set, where the choice of projection set differentiates SAEs (see Fig. 2). We formalize such nonlinearities as projection nonlinearities, as (re)defined below.

**Definition D.1** (Projection Nonlinearity). Let $\boldsymbol{v} \in \mathbb{R}^s$ be a pre-activation vector. A projection nonlinearity $\boldsymbol{\Pi}_{\mathcal{S}} \{\cdot\} : \mathbb{R}^s \to \mathbb{R}^s$ is defined as:

$$\boldsymbol{\Pi}_{\mathcal{S}} \{\boldsymbol{v}\} = \arg\min_{\boldsymbol{\pi} \in \mathcal{S}} \|\boldsymbol{\pi} - \boldsymbol{v}\|_2^2, \tag{12}$$

where $\mathcal{S} \subseteq \mathbb{R}^s$ is the constraint set onto which $\boldsymbol{v}$ is orthogonally projected. The structure of $\mathcal{S}$ determines the properties of the nonlinearity.

We will say a function $\boldsymbol{f}(\cdot)$ is a **Projection Encoder** if it uses a projection nonlinearity $\boldsymbol{g}(\cdot)$ applied to a linear transformation of the input. This is equivalent to using $\boldsymbol{v} = \boldsymbol{W}^\top \boldsymbol{x} + \boldsymbol{b}_e$, and $\boldsymbol{f} = \boldsymbol{g}(\boldsymbol{v})$ (see Eq. 2), where $\boldsymbol{g}$ is a projection nonlinearity. Popular SAEs can be understood as a similar Projection Encoder with different projection nonlinearities, as shown in Tab. 4 (see Theorem D.3 for a derivation).

Table 4: **Projection Nonlinearities in SAE Encoders.** Each model can be understood by its nonlinear orthogonal projection $\boldsymbol{g}(\cdot)$ onto a constraint set $\mathcal{S}$ which determines its activation behavior, sparsity structure, and implicit data assumptions.

| Model | $\boldsymbol{g}(\boldsymbol{v})$ |
|---|---|
| ReLU | $\boldsymbol{\Pi}_{\mathcal{S}} \{\boldsymbol{v}\}, \mathcal{S} = \{\boldsymbol{x} \in \mathbb{R}^s : \boldsymbol{x} \geq \boldsymbol{0}\}$ |
| TopK | $\boldsymbol{\Pi}_{\mathcal{S}} \{\boldsymbol{v}\}, \mathcal{S} = \{\boldsymbol{x} \in \mathbb{R}^s : \boldsymbol{x} \geq \boldsymbol{0}, \|\boldsymbol{x}\|_0 \leq k\}$ |
| Heaviside ($H$) | $\boldsymbol{\Pi}_{\mathcal{S}} \{\boldsymbol{v} + \frac{1}{2}\boldsymbol{1}\}, \mathcal{S} = \{0, 1\}^s$ |
| JumpReLU | $\text{ReLU}(\boldsymbol{v} - \boldsymbol{\theta}) + \boldsymbol{\theta} \odot H(\boldsymbol{v} - \boldsymbol{\theta})$ |

**Lemma D.2** (Elementwise projections). *For projection nonlinearities whose projection sets satisfy componentwise constraints, i.e.* $\mathcal{S} = \{\boldsymbol{x} \in \mathbb{R}^s : f(x_j) \leq 0, h(x_k) = 0 \forall j, k \in [s]\}$, *the projection problem can be decoupled and broken down into a combination of elementwise projections, leading to an elementwise nonlinearity. The converse is also true: any elementwise nonlinearity which is also a projection nonlinearity can be written as a combination of elementwise projections, leading to componentwise constraints on the projection set*

*Proof.*

$$\boldsymbol{\Pi}_{\mathcal{S}} \{\boldsymbol{x}\} = \arg\min_{\boldsymbol{\pi} \in \mathcal{S}} \|\boldsymbol{\pi} - \boldsymbol{x}\|^2 \tag{13}$$

$$= \arg\min_{f(\pi_j) \leq 0, g(\pi_j) = 0, j \in [s]} \sum_k (\pi_k - x_k)^2 \tag{14}$$

$$= (..., \arg\min_{f(\pi_k) \leq 0, g(\pi_k) = 0,} (\pi_k - x_k)^2, ...) \tag{15}$$

$$\text{i.e.,} \quad \boldsymbol{\Pi}_{\mathcal{S}} \{\boldsymbol{x}\}_k = \arg\min_{f(\pi_k) \leq 0, g(\pi_k) = 0,} (\pi_k - x_k)^2 \tag{16}$$

This is a consequence of the objective function above (squared euclidean norm of the difference $\boldsymbol{\pi} - \boldsymbol{x}$) decomposing into a sum over componentwise functions. The above argument can be traced backward, since all steps are invertible, which proves the converse. $\square$

**Theorem D.3** (Projection Nonlinearities). *ReLU, TopK, JumpReLU are simple combinations of orthogonal projections onto nonlinearity-specific sets: ReLU is a projection onto the positive orthant, TopK is a projection onto the union of all k-sparse subspaces, and JumpReLU is a sum of shifted ReLU and shifted Heaviside step, which itself is a projection onto the corners of a hypercube.*

*Proof.* First consider the ReLU nonlinearity, defined for $\boldsymbol{x} \in \mathbb{R}^s$ as:

$$\boldsymbol{z} = \text{ReLU}(\boldsymbol{x}) \tag{17}$$

$$z_i = \begin{cases} x_i & \text{if } x_i \geq 0 \\ 0 & \text{else} \end{cases} \tag{18}$$

This is an elementwise nonlinearity, so it suffices to show that each component can be written as a projection ( from Lemma D.2). Consider this reformulation:

$$z_i = \arg\min_{\pi_i \geq 0} (x_i - \pi_i)^2 \tag{19}$$

This is equivalent to ReLU, since for all non-negative values, it equals the input, while it is 0 (nearest non-negative point) for all negative inputs. Using Lemma D.2, ReLU is a projection nonlinearity with projection set $\mathcal{S} = \{\boldsymbol{x} \in \mathbb{R}^s : x_i \geq 0 \forall i \in [s]\}$.

JumpReLU is defined as:

$$\text{JumpReLU}(\boldsymbol{x}) = \boldsymbol{x} \odot \mathbb{H}(\boldsymbol{x} - \boldsymbol{\theta}) \tag{20}$$

$$= (\boldsymbol{x} - \boldsymbol{\theta} + \boldsymbol{\theta}) \odot \mathbb{H}(x - \theta) \tag{21}$$

$$= \text{ReLU}(\boldsymbol{x} - \boldsymbol{\theta}) + \boldsymbol{\theta} \odot \mathbb{H}(\boldsymbol{x} - \boldsymbol{\theta}) \tag{22}$$

where the heaviside step function $\mathbb{H}$ is:

$$\mathbb{H}(\boldsymbol{x}) = \mathbb{I}(\boldsymbol{x} > 0) \tag{23}$$

which is performed elementwise. Thus, JumpReLU (and the heaviside step) is also an elementwise nonlinearity. Consider the step function:

$$\mathbb{H}(\boldsymbol{x})_i = \mathbb{H}(x_i) = \begin{cases} 1 & \text{if } x_i \geq 0 \\ 0 & \text{else} \end{cases} \tag{24}$$

$$= \underset{\pi_i \in \{0,1\}}{\arg\min}(x_i + 0.5 - \pi_i)^2 \tag{25}$$

which is a shifted version of a projection. Again using Lemma D.2, $\mathbb{H}$ is a projection nonlinearity with projection set $\mathcal{S} = \{\boldsymbol{x} \in \mathbb{R}^s : x_i \in \{0, 1\}\}$, i.e., the corners of a unit hypercube.

The TopK nonlinearity is defined as:

$$y_j = \text{ReLU}(x_j) \tag{26}$$

$$\text{TopK}(\boldsymbol{x})_j = y_j \, \mathbb{I}\big(y_j \geq y_p \forall p \in \mathcal{M} : |\mathcal{M}| = s - K\big) \tag{27}$$

where $s$ is the dimension of the space. Note that topK typically includes a ReLU applied first ([13]), making all entries of the vector non-negative followed by choosing the $k$-largest entries of $ReLU(\boldsymbol{x})$. Consider a projection onto the union of all $k$-dimensional axis-aligned subspaces. With non-negative entries (due to ReLU), this would lead to choosing the k largest entries of $x$:

$$\underset{\pi: \, \pi \text{ is } k-\text{sparse}}{\arg\min} \|x - \pi\|_2^2 = \underset{\boldsymbol{\pi}: \, \boldsymbol{\pi} \text{ is } k-\text{sparse}}{\arg\min} \sum_i (x_i - \pi_i)^2 \tag{28}$$

$$= TopK(\boldsymbol{x}) \tag{29}$$

This completes the proof. □

**Theorem D.4.** *Projection nonlinearities satisfy the following properties:*

1. *For points within the set $\mathcal{S}$, projection is an identity map*
$$\boldsymbol{x} \in \mathcal{S} \implies \boldsymbol{\Pi}_{\mathcal{S}}\{\boldsymbol{x}\} = \boldsymbol{x}$$

2. *For points outside the set $\mathcal{S}$, projection is onto the boundary*
$$\boldsymbol{x} \notin \mathcal{S} \implies \boldsymbol{\Pi}_{\mathcal{S}}\{\boldsymbol{x}\} \in \partial\mathcal{S}$$

3. *If $\partial\mathcal{S}$ is a flat (linear manifold), or a subset of a flat (with flat boundaries), projection of points outside the set $\mathcal{S}$ is either piecewise linear or constant:*
$$\boldsymbol{\Pi}_{\mathcal{S}}\{\alpha\boldsymbol{x}_1 + \beta\boldsymbol{x}_2\} = \alpha\boldsymbol{\Pi}_{\mathcal{S}}\{\boldsymbol{x}_1\} + \beta\boldsymbol{\Pi}_{\mathcal{S}}\{\boldsymbol{x}_2\} \ \text{ for } \alpha, \beta \in \mathcal{T}, \ \text{OR}$$
$$\boldsymbol{\Pi}_{\mathcal{S}}\{\boldsymbol{x}\} = \boldsymbol{c}, \ \boldsymbol{x} \in \boldsymbol{D} \text{ (a linear piece)}$$
*where $\boldsymbol{x}_1, \boldsymbol{x}_2 \notin \mathcal{S}$, $\mathcal{T} \subseteq \mathbb{R}$ is suitably defined to confine $\boldsymbol{x}$ to the corresponding linear piece*

*Proof.* (sketch) (1) is trivial and follows from the definition of projection nonlinearities (Eq. 3). For (2), suppose $\boldsymbol{\Pi}_{\mathcal{S}}\{\boldsymbol{x}\}$ is in the interior of $\mathcal{S}$. This implies that $\exists \boldsymbol{y} \in Int(\mathcal{S})$ such that $\boldsymbol{y} = \alpha\boldsymbol{x} + (1-\alpha)\boldsymbol{\Pi}_{\mathcal{S}}\{\boldsymbol{x}\}, \alpha \in (0, 1]$ and therefore $\|\boldsymbol{y}-\boldsymbol{x}\|^2 < \|\boldsymbol{x}-\boldsymbol{\Pi}_{\mathcal{S}}\{\boldsymbol{x}\}\|^2$, which is a contradiction. Thus $\boldsymbol{\Pi}_{\mathcal{S}}\{\boldsymbol{x}\} \in \partial\mathcal{S}$.
For (3), one can consider the section of the boundary $\partial\mathcal{S}$ that is closest to $\boldsymbol{x}$, and extend it to form a subspace (possible since it is flat). Since projections onto subspaces are linear operations, $\boldsymbol{\Pi}_{\mathcal{S}}\{\boldsymbol{x}\}$ is linear in some neighborhood, and thus piecewise linear. In some cases, there is a single *corner* point of $\mathcal{S}$ that is closest to $\boldsymbol{x}$, in which case the projection is a constant. □

Projection nonlinearities are orthogonal projections onto various sets. For points within the set $\mathcal{S}$, projection is the point itself, while for points outside, the projection is onto the boundary $\partial\mathcal{S}$ (Theorem D.4 in Appendix). For projections to be well defined everywhere, the set $\mathcal{S}$ must be closed (so that the boundary belongs to the set, i.e., $\partial\mathcal{S} \in \mathcal{S}$). Note that if the set $\mathcal{S}$ is a subspace of $\mathbb{R}^s$, projection is a linear map. Therefore, the nonlinearity of projection nonlinearities comes from choosing either a subset of a subspace, or a non-flat manifold. Sparsity in projection nonlinearities is a consequence of the projection set having edges/corners along sparse subspaces.

## D.2 Receptive fields of various SAEs

First, we (re)define the four SAE encoders we study in this section:

$$\text{ReLU SAE: } \boldsymbol{z} = \text{ReLU}(\boldsymbol{W}^T\boldsymbol{x} + \boldsymbol{b}) \tag{30}$$

$$\text{JumpReLU SAE: } \boldsymbol{z} = \text{JumpReLU}(\boldsymbol{W}^T\boldsymbol{x} + \boldsymbol{b}) \tag{31}$$

$$\text{TopK SAE: } \boldsymbol{z} = \text{TopK}(\boldsymbol{W}^T\boldsymbol{x}) \tag{32}$$

$$\text{SpaDE: } \boldsymbol{z} = \text{Sparsemax}(-\lambda d(\boldsymbol{x}, \boldsymbol{W})) \tag{33}$$

$$d(\boldsymbol{x}, \boldsymbol{W})_i = \|\boldsymbol{x} - \boldsymbol{w}_i\|_2^2 \tag{34}$$

This section discusses the piecewise linear (affine) regions (by showing that each of the above is a piecewise linear function) and neuron receptive fields in input space for each of the four SAEs (ReLU, JumpReLU, TopK, SpaDE). Projection nonlinearities become piecewise linear when the projection sets have flat faces. Under the requirement of monosemanticity, the structure of receptive fields directly implies the assumption that concepts in data have the same structure as the receptive field.

**Intuition behind Theorem 4.1**  Receptive fields for ReLU and JumpReLU SAEs are half-spaces because of the linear transform in the encoder, which needs to be positive for the SAE latent to be active (due to the ReLU/ JumpReLU nonlinearity). In TopK SAE, for a given latent to be active, its linear transform in the encoder must be non-negative (since a ReLU is used in TopK SAE implementation [13]), and must exceed at least $s - K$ other linear transforms (where $s$ is the SAE width). This gives us an intersection of multiple half-spaces through the origin, leading to hyperpyramids and thus angular separation.

For projection-based encoders, the receptive field can be rewritten as

$$\mathcal{F}_k = \boldsymbol{f}^{-1}\big(\mathcal{S} \cap \{z_k > 0\}\big),$$

where $\mathcal{S}$ is the projection set of the encoder.

That is, $\mathcal{F}_k$ is the pre-image of the intersection of the projection set with the half-space $\{z_k > 0\}$. Alternatively, it can be viewed as the complement of the pre-image of the set $\mathcal{S} \cap \{z_k = 0\}$, where the hyperplane $z_k = 0$ indicates latent $k$ is "dead". This expression shows the explicit relation between the projection set and the receptive field properties of the SAE.

First note that all four nonlinearities have some level of sparsity, i.e., some neurons are *turned off* at times. The following observation is crucial in formulating the piecewise linear regions:

**Lemma D.5** (Gating). *Given the indices $\mathcal{M} = \{i_1, i_2, ..., i_{|\mathcal{M}|}\}$ of active neurons (with nonzero outputs), ReLU, JumpReLU, TopK and Sparsemax are all affine functions of their inputs.*

Lemma D.5 indicates that the nonlinearity in these transformation lies only in their *gating*, or selection of active indices. Thus, each linear (affine) region is characterized by a specific choice of indices $\mathcal{M}$ of active neurons. Note that not all choices of indices may be allowed by the nonlinearity. Denote the set of allowed indices by $\mathbb{M}$.

Let $\mathcal{L}_{\mathcal{M}} \subseteq \mathbb{R}^n$ denote the piecewise linear (affine) region corresponding to active indices $\mathcal{M}$.

**Lemma D.6.** *The set $\{\mathcal{L}_{\mathcal{M}} : \mathcal{M} \in \mathbb{M}\}$ of all piecewise linear regions forms a partition of $\mathbb{R}^n$.*

Using the Gating lemma, we can associate each set of active indices to a piecewise linear region, and identify receptive fields as unions of such piecewise linear regions.

**Lemma D.7** (Receptive fields and piecewise linear regions). *A neuron's receptive field is a union of piecewise linear regions where the neuron is active:*

$$\mathcal{F}_k = \cup_{\mathcal{M}:k\in\mathcal{M}}\mathcal{L}_{\mathcal{M}}$$

We now use the above results and obtain the piecewise linear regions for each of the four SAEs defined previously.

### D.2.1 ReLU, JumpReLU SAE

First note that the piecewise linear regions and receptive fields of ReLU and JumpReLU SAEs are the same—since in both cases, the gating appears through the heaviside step function ($ReLU(\boldsymbol{x}) = \boldsymbol{x} \odot \mathbb{I}(\boldsymbol{x} \geq 0)$). Thus, we develop the linear pieces and receptive fields only for ReLU, since the corresponding ones for JumpReLU are identical. The piecewise linear regions of latents in ReLU SAE are described by the following claim:

**Claim D.1.** *For a layer defined as in Eq. 30, $\mathcal{L}_{\mathcal{M}}$ is given as:*

$$\mathcal{L}_{\mathcal{M}} = \{\boldsymbol{x} \in \mathbb{R}^n : \boldsymbol{w}_m^T x + b_m \geq 0 \forall m \in \mathcal{M}, \boldsymbol{w}_q^T x + b_q < 0 \forall q \notin \mathcal{M}\} \tag{35}$$

*Thus, $\mathcal{M}$ is an intersection of N half-spaces, and thus is a convex polytope which may be bounded or unbounded.*

*Proof.* This is a consequence of the observation in Lemma D.5 and the definition of the relu model 30. $\qquad\square$

**Lemma D.8.** *If $b = 0$ in Eq. 30, then $\mathcal{L}_{\mathcal{M}}$ are unbounded convex polytopes with only one corner at the origin and flat faces, i.e., they are (unbounded) hyperpyramids.*

Thus, bias plays an important role in ReLU layers, allowing piecewise linear regions that are convex polytopes with multiple corners anywhere in space. The greater flexibility in defining the pieces allows greater expressivity by capturing a larger class of functions. The following (somewhat obvious) claim describes the receptive fields of model 1 neurons.

**Claim D.2.** *In Model 1 (30), for a given neuron $k \in [n]$, the receptive field $\mathcal{F}_k$ is given as:*

$$\mathcal{F}_k = \{\boldsymbol{x} \in \mathbb{R}^n : \boldsymbol{w}_k^T \boldsymbol{x} + b_k \geq 0\} \tag{36}$$

*which is a half-space defined by the normal vector $\boldsymbol{w}_k$ and bias $b_k$.*

This is a straightforward consequence of the definition of the ReLU model in Eq. 30.

### D.2.2 TopK SAE

**Claim D.3.** *For a layer defined as in Eq. 32, $\mathcal{L}_{\mathcal{M}}$ is given as:*

$$\mathcal{L}_{\mathcal{M}} = \{\boldsymbol{x} \in \mathbb{R}^n : \boldsymbol{w}_m^T \boldsymbol{x} \geq \boldsymbol{w}_q^T \boldsymbol{x} \forall m \in \mathcal{M}, q \notin \mathcal{M}\} \tag{37}$$

*Thus, $\mathcal{M}$ is an intersection of $K(N-K)$ half-spaces all passing through the origin, and thus is a convex polytope which may be bounded or unbounded. In fact, it is an unbounded hyperpyramid, with a corner at the origin and flat faces. The normals to these half-spaces are pairwise differences between active and inactive weight vectors.*

This again follows from the Gating Lemma D.5.

**Claim D.4.** *In Model 2 (32), for a given neuron $k \in [n]$, the receptive field $\mathcal{F}_k$ is given as:*

$$\mathcal{F}_k = \cup_{\mathcal{M}:k\in\mathcal{M}}\mathcal{L}_{\mathcal{M}} \tag{38}$$

*which is a union of hyperpyramids with a corner at the origin. Note that in typical implementations of TopK, a pre-encoder bias is included, so the corner of the hyperpyramids is at the pre-encoder bias.*

### D.2.3 SpaDE

**Claim D.5.** *For a layer defined as in Eq. 5, $\mathcal{L}_{\mathcal{M}}$ is given as:*

$$\mathcal{L}_{\mathcal{M}} = \left\{ \boldsymbol{x} \in \mathbb{R}^n : \|\boldsymbol{x} - \boldsymbol{w}_m\|_2^2 - \frac{1}{|\mathcal{M}|}\sum_{j\in\mathcal{M}}\|\boldsymbol{x} - \boldsymbol{w}_j\|_2^2 - \frac{1}{\lambda|\mathcal{M}|} \begin{cases} \leq 0, & if\, m \in \mathcal{M} \\ > 0, & m \notin \mathcal{M} \end{cases} \right\} \tag{39}$$

$$= \left\{ \boldsymbol{x} \in \mathbb{R}^n : \right. \tag{40}$$

$$\left. \Big(\boldsymbol{w}_m^T - \frac{1}{|\mathcal{M}|}\sum_{j\in\mathcal{M}}\boldsymbol{w}_j^T\Big)\boldsymbol{x} - \Big(\|\boldsymbol{w}_m\|_2^2 - \frac{1}{|\mathcal{M}|}\sum_{j\in\mathcal{M}}\|\boldsymbol{w}_j\|_2^2\Big) + \frac{1}{\lambda|\mathcal{M}|} \begin{cases} \geq 0, & if\, m \in \mathcal{M} \\ < 0, & m \notin \mathcal{M} \end{cases} \right\}$$
$$\tag{41}$$

*Thus, $\mathcal{M}$ is an intersection of $N$ half-spaces, and thus a convex polytope. Note that the normal to each half space is now chosen in an input-adaptive fashion ($m \in \mathcal{M}$) and is locally centered using the mean of other nearby prototypes that are active, i.e., $\left( \boldsymbol{w}_m^T - \frac{1}{|\mathcal{M}|} \sum_{j \in \mathcal{M}} \boldsymbol{w}_j^T \right)$ where $\mathcal{M}$ is input adaptive. An alternate interpretation is using the first equation above, which defines the region as the set of points whose distance to active prototypes is within a tolerance of the average distance to all active prototypes, while distance to inactive prototypes is larger than the average distance to active prototypes.*

*Proof.* This is again a consequence of the definition of sparsemax [44]. $\qquad \square$

**Claim D.6.** *In SpaDE (33), for a given neuron $k \in [n]$, the receptive field $\mathcal{F}_k$ is given as:*

$$\mathcal{F}_k = \cup_{\mathcal{M}:k \in \mathcal{M}} \mathcal{L}_\mathcal{M} \tag{42}$$

*which is a union of convex polytopes, each of which includes the latent $k$ in the set of active indices $\mathcal{M}$. Due to the use of euclidean distances in choosing active indices, the receptive field is a union of convex polytopes in the vicinity of the prototype $a_k$ of latent $k$. This incorporates the notion of locality and flexibility in receptive field shapes, allowing latents to capture nonlinearly separable concepts.*

## D.3 KDS and Sparse Coding

K-Deep Simplex (KDS) [45] is the sparse coding framework which forms the outer optimization in the SpaDE. While this is a different framework, in this section we show that it is general enough to capture the standard sparse coding, i.e., for data generated using standard sparse coding, there exists a corresponding KDS framework that could have generated the same data. Note that we may have to increase the latent dimension (number of dictionary atoms) by one to obtain the corresponding KDS framework. This is stated and proved (with a constructive proof) in the following theorem.

**Theorem D.9** (KDS can capture standard sparse coding). *Given data $\mathcal{D} = \{\boldsymbol{x}^{(1)}, ..., \boldsymbol{x}^{(P)}\}$ generated from a standard sparse coding generative model, i.e., $\boldsymbol{x} = \boldsymbol{D}\boldsymbol{z} + \eta$, where dictionary atoms (columns of $\boldsymbol{D}$) have unit norm and $\boldsymbol{z}$ is unconstrained, there exists a scaling of the data such that it can be represented using the K-Deep Simplex [45] framework, i.e., $\tilde{\boldsymbol{x}} = \kappa \boldsymbol{x} = \tilde{\boldsymbol{D}}\tilde{\boldsymbol{z}} + \tilde{\eta}$, where $\tilde{\boldsymbol{z}} \in \Delta^s$.*

*Proof.* Consider the following scalar:

$$\kappa = \left( \max_{\boldsymbol{x} \in \mathcal{D}} \sum_i z_i(\boldsymbol{x}) \right)^{-1}$$

Normalizing data using $\kappa$ above gives us,

$$\tilde{\boldsymbol{x}} = \kappa \boldsymbol{x}$$
$$= \boldsymbol{D} \frac{\boldsymbol{z}}{\max_{\boldsymbol{x} \in \mathcal{D}} \sum_i z_i(\boldsymbol{x})} + \kappa \eta$$
$$= \boldsymbol{D}\hat{\boldsymbol{z}} + \tilde{\eta}$$

By definition, $\hat{\boldsymbol{z}}$ defined above always satisfies $\sum_i \hat{z}_i \leq 1$, so let $\beta = 1 - \sum_i \hat{z}_i$. Appending an all-zeros dictionary atom to $\boldsymbol{D}$, $\tilde{\boldsymbol{D}} = [\boldsymbol{D}, \boldsymbol{0}]$ and assigning the residual to $\tilde{\boldsymbol{z}} = [\hat{\boldsymbol{z}}^T, \beta]^T$ gives us the following:

$$\tilde{\boldsymbol{x}} = \tilde{\boldsymbol{D}}\tilde{\boldsymbol{z}} + \tilde{\eta}, \quad \text{where } \tilde{\boldsymbol{z}} \in \Delta^s$$

implying that the original data can be represented in the framework of KDS. $\qquad \square$

## D.4 SpaDE

Sparsemax is a projection onto the probability simplex, which can be written as (see Proposition 1 in [44])

$$\text{Let } \boldsymbol{z} = \text{Sparsemax}(\boldsymbol{y}) \tag{43}$$

$$\text{Then, } z_i = \text{ReLU}(y_i - \frac{1}{|\mathcal{M}|} \sum_{j \in \mathcal{M}} y_j + \frac{1}{|\mathcal{M}|}) \tag{44}$$

SpaDE is defined using squared euclidean distances between an input vector and some *prototypes* (or landmarks) in input space (Eq. 5), which gives us

$$y_i = -\lambda |\boldsymbol{x} - \boldsymbol{w}_i|_2^2 \tag{45}$$

$$\implies \text{Sparsemax}(\boldsymbol{y})_i = \text{ReLU}\left(2\lambda(\boldsymbol{w}_i - \frac{1}{|\mathcal{M}|}\sum_j \boldsymbol{w}_j)^T \boldsymbol{x} - \lambda(|\boldsymbol{w}_i|^2 - \frac{1}{|\mathcal{M}|}\sum_j |\boldsymbol{w}_j|^2) + \frac{1}{|\mathcal{M}|}\right) \tag{46}$$

$$= ReLU(\tilde{\boldsymbol{W}}(\boldsymbol{x})\boldsymbol{x} + \tilde{\boldsymbol{b}}_e(\boldsymbol{x})) \tag{47}$$

where $\tilde{\boldsymbol{W}}(\boldsymbol{x}) = 2\lambda(\boldsymbol{w}_i - \frac{1}{|\mathcal{M}|}\sum_j \boldsymbol{w}_j)$, $\tilde{\boldsymbol{b}}_e = -\lambda(|\boldsymbol{w}_i|^2 - \frac{1}{|\mathcal{M}|}\sum_j |\boldsymbol{w}_j|^2) + \frac{1}{|\mathcal{M}|}$ and $\mathcal{M}$ is the set of active indices, which is uniquely determined by the constraint $\sum_i \text{Sparsemax}(\boldsymbol{y})_i = 1$ (see Proposition 1 in [44] for uniqueness). Note that $\tilde{\boldsymbol{W}}(\boldsymbol{x}), \tilde{\boldsymbol{b}}_e(\boldsymbol{x})$ are both *piecewise constant* on regions of input space marked by the same choice of active indices.

Thus, SpaDE is equivalent to a ReLU SAE, but with a linear transformation and bias that are input-adaptive (piecewise constant). SpaDE is thus piecewise linear and continuous (continuity follows from the continuity of sparsemax). Note that this is a nontrivial result: despite appearing quadratic in input due to the use of squared euclidean distances, SpaDE is a piecewise linear function of the input. This result is also exact, and is NOT a first order Taylor series approximation.

However, SpaDE differs from a ReLU SAE by using linear transformations defined with respect to a local origin, which is uniquely determined by the set of active SAE latents, similar to recent work on steering [80].

Since SAEs are completely described by their inner and outer optimization problems (see Theorem 3.1), we now describe these components for SpaDE.

The inner optimization (Eq. 4) for the SpaDE is as follows:

$$\boldsymbol{F}(\boldsymbol{\pi}, \boldsymbol{W}, \boldsymbol{x}) = \sum_i \pi_i \|\boldsymbol{x} - \boldsymbol{w}_i\|_2^2 + \frac{1}{2\lambda}\|\boldsymbol{\pi}\|_2^2$$

$$\mathcal{S} = \{\boldsymbol{\pi} \in \mathbb{R}^s : \ \pi_i \geq 0, \sum_i \pi_i = 1\} \tag{48}$$

This resembles one-sided optimal transport with a squared 2-norm regularizer. This problem is one-sided because there is no constraint on how much *weight* sits on each prototype across different inputs (optimization is performed independently for each input). The squared $2-$norm regularizer is known to lead to sparse transport plans in the optimal transport literature (see [81]).

The outer optimization for SpaDE (Eq. 4) is a locality-enforced version of dictionary learning called K-Deep Simplex (KDS) [45]. In this framework, the sparse code is constrained to belong to the probability simplex, i.e., $\boldsymbol{z} \in \Delta^s = \{\boldsymbol{y} \in \mathbb{R}^s : \sum_i y_i = 1, y_i \geq 0 \forall i\}$, while the dictionary atoms $\boldsymbol{D}$ are unconstrained. The distance-weighted L1 regularizer encourages each datapoint to use those dictionary atoms which are close to itself in euclidean distance, inducing a soft clustering bias. Even though this is a different dictionary learning framework than standard sparse coding, it is expressive enough to capture the standard sparse coding setup, i.e., for any standard sparse coding problem, there exists an equivalent KDS problem (see Theorem D.9 in Appendix).

While this outer optimization (KDS) is a different problem than the standard dictionary learning problem, it may be useful for interpretability since it has the following advantages:

1. It avoids shrinkage, since the L1 norm of the sparse representation $\boldsymbol{z}(\boldsymbol{x})$ is constrained to equal 1 for all inputs
2. Constraining the sparse code to the probability simplex finds support in an oft-cited paper demonstrating the linear representation hypothesis in word embeddings under a random-walk based generative model of language [9]. Their main result (Theorem 2) shows that representations are *convex* combinations of concepts, as opposed to unconstrained linear combinations, which is better interpreted as assigning vectors (with magnitude and direction; alternatively, locations) to concepts rather than directions (without magnitude). This idea of concepts as vectors has also been demonstrated both theoretically and empirically in the final layer representations of language models [82].

Note how SpaDE satisfies the two data properties of nonlinear separability and heterogeneity:

1. The projection set $\mathcal{S}$ in SpaDE is the probability simplex, which admits edges/corners with varying levels of sparsity, thereby allowing the representation of heterogeneous concepts. For any choice of $k \in \{1, 2, ..., s\}$, there are $\binom{s}{k}$ choices of indices $\mathcal{M}_k$ for a $k$-sparse representation, and points $\{\boldsymbol{x} \in \mathbb{R}^s : x_i = 0, i \notin \mathcal{M}_k, \sum_{j \in \mathcal{M}_k} x_j = 1, x_j \geq 0\} \subseteq \Delta^s$ which admit this level of sparsity, thereby capturing concept heterogeneity.

2. The receptive fields of SpaDE (see App. D.2.3) are local to each prototype (encoder weight vector), and are flexibly defined as the union of convex polytopes. This allows latents in SpaDE to become monosemantic to concepts which are nonlinearly separable from the rest of the data.

# E   Further Results

In this section, we present a more detailed analysis of the results from each of our four experiments.

## E.1   Separability Experiment

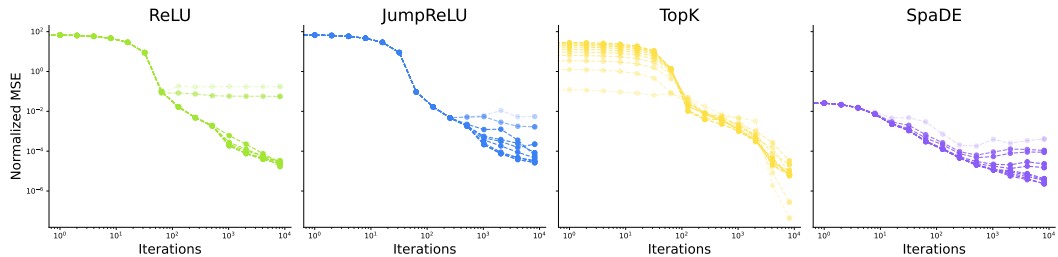

Figure E.1: Evolution of normalized MSE with training iterations for various SAEs on the *separability experiment*. Color intensity is proportional to $L_0$ (darker colors imply more dense SAE latents).

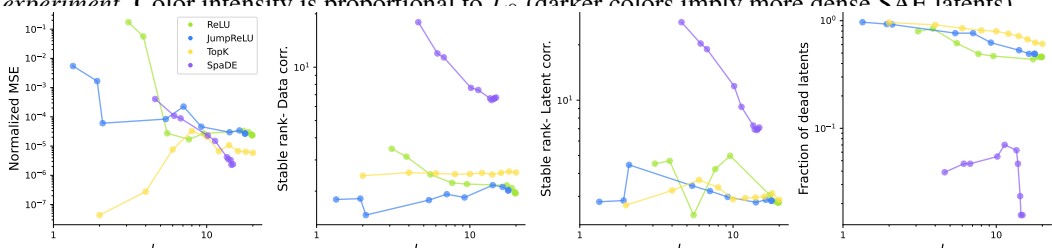

Figure E.2: Normalized MSE (normalized with variance of data), Stable ranks (of data correlations, latent correlations matrices), and fraction of dead latents as a function of sparsity ($L_0$) for the *separability experiment* (Sec. 5.1)

Fig. E.1 shows the evolution of normalized MSE (NMSE- MSE normalized by the variance of data) with training iterations for each SAE, for different levels of sparsity. Note that denser representations (higher $L_0$ and thus darker colors in Fig. E.1) lead to lower NMSE. While all SAEs end up at similar levels of NMSE, their ability to extract concepts from data is markedly different (as described in Sec. 4). A per-concept breakdown of training dynamics is shown in Fig. E.3. For comparison, this figure also includes the mean of the squared norm of each concept (which equals MSE if the SAE predicts the origin for all inputs), variance of each concept (which equals MSE if the SAE predicts the mean of each concept). Thus, SAEs whose MSE saturates at the concept variance are likely to be predicting the mean of the concept for all points, whereas when MSE goes below concept variance, the SAE explains within-concept variance. Also shown in gray is MSE with respect to the center of each concept, which ideally must match concept variance if the SAE reconstructs all points (which is observed in most cases).

In Fig. E.2, final NMSE as a function of sparsity ($L_0$) shows that while all SAEs have comparable MSE-sparsity curves at dense representations (high $L_0$), TopK's NMSE goes down significantly more than others. This is a consequence of TopK learning a redundant solution, by just using two latents as an orthogonal basis to represent all data. Fraction of dead latents show large numbers of dead latents at high sparsity levels for ReLU, JumpReLU and TopK, with this going down (exponentially)

as representations become more dense. However, SpaDE shows significantly fewer dead latents at all levels of sparsity. Stable ranks of cosine similarities between latent representations of pairs of data points (data corr.), and between pairs of latents across all data points (latent corr.) show that SpaDE has very high stable ranks, indicating high specialization of latents. The other SAEs have comparable stable ranks, all much lower than the desirable stable rank of 6 (equal to the number of clusters in data).

The SAE latent activation profiles for each concept are shown as histograms in Fig. E.4. While variations exist across concepts, there is a common structure to the profiles for each SAE (SpaDE appears *pointy*, indicating a second mode other than zero).

Cosine similarities between latent representations of pairs of data points are shown for different levels of sparsity in Fig. E.5. Notice that SpaDE has the lowest cross-concept correlations of all SAEs, and these correlations do not decrease much especially in ReLU and JumpReLU. The corresponding figure with similarities between pairs of latents across all datapoints is in Fig. E.6. Here, the number of dead latents increases with increasing sparsity, leading to very few active latents (only active latents are shown). Broadly, note the decrease in co-occurrences with increase in sparsity- also note how ReLU and JumpReLU result in newer correlation structures with greater sparsity.

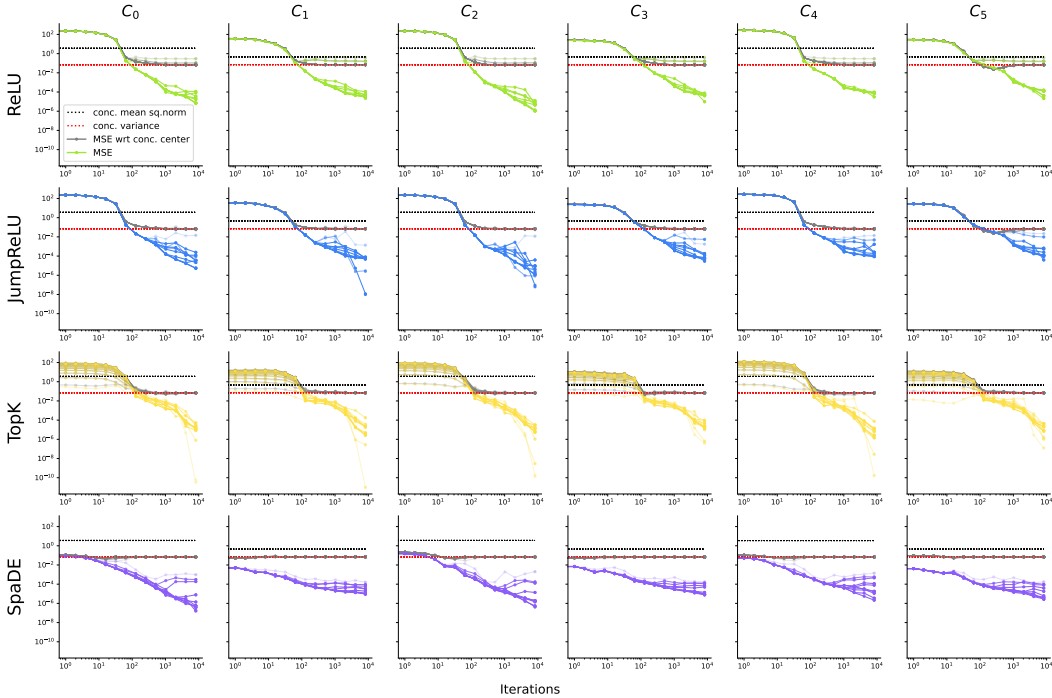

Figure E.3: Training dynamics for each concept (column) across SAEs (rows) for *separability experiment*: colored solid lines are MSE, with intensity of color proportional to $L_0$. Gray lines show MSE of SAE predictions with respect to the center of each cluster; intensity is again proportional to $L_0$. . Black dotted line shows the mean squared norm of each cluster, which would equal the MSE if the SAE predicted the origin for all datapoints. Red dotted line shows variance of each cluster, which again equals MSE if an SAE predicts the center of the cluster. Note that when a model reconstructs data well, MSE wrt cluster center equals the variance of the cluster (as observed here)

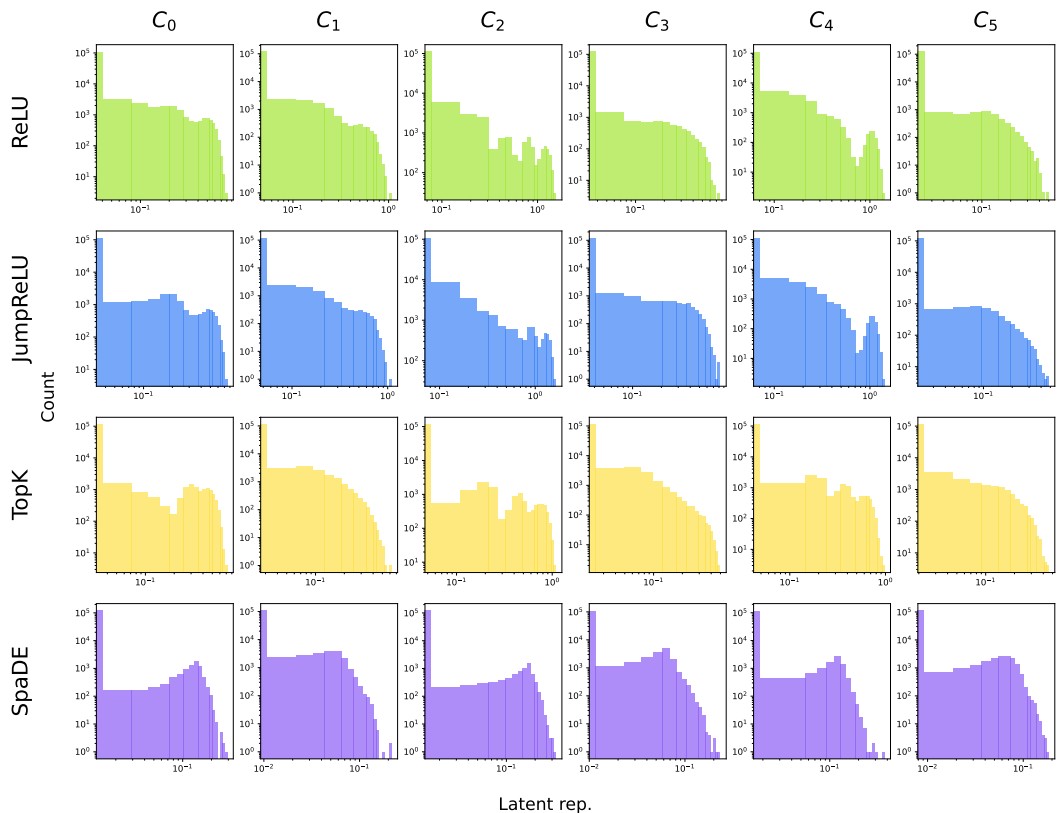

Figure E.4: Histogram of latent representations for each concept of various SAEs on the *separability experiment*.

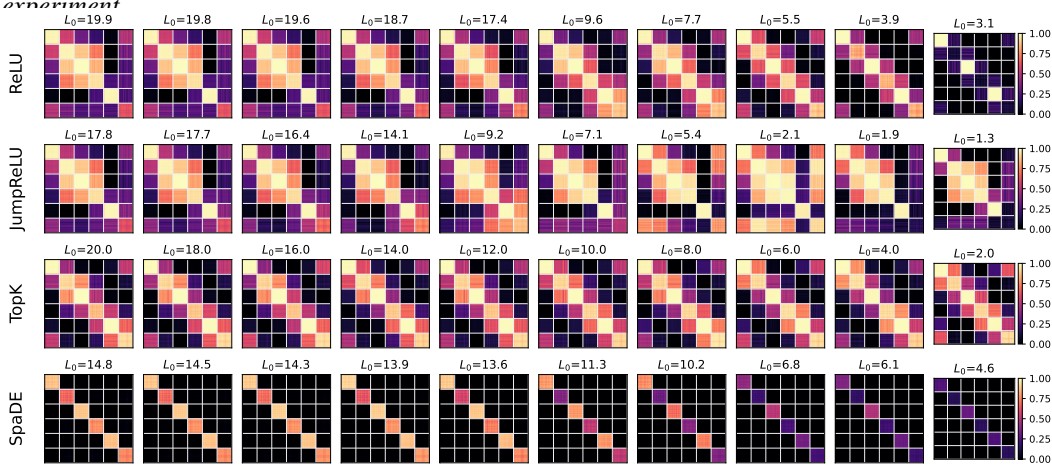

Figure E.5: Data correlations for various sparsity levels on the *separability experiment*: Pairwise cosine similarities between SAE latent representations of datapoints. White lines separate different concepts.

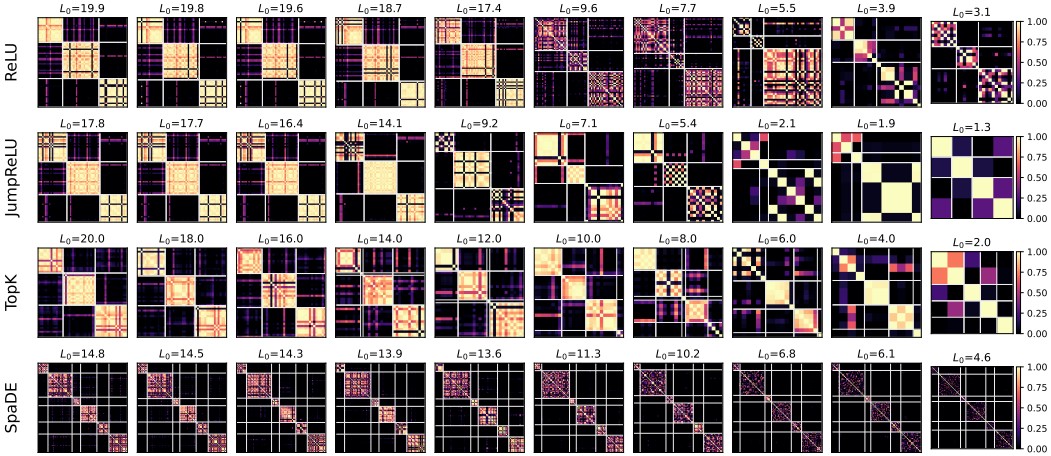

Figure E.6: Latent correlations for various sparsity levels on the *separability experiment*: Pairwise cosine similarities: pairwise cosine similarities between different SAE latents, computed across data from all concepts.

## E.2  Heterogeneity Experiment

The overall training dynamics (on data from all concepts) is shown in Fig. E.8- note, again, that for low sparsity (high $L_0$, darker color) all SAEs reach similar levels of NMSE, but differ for higher sparsity levels. The per-concept breakdown of MSE, and comparison with mean squared norm, concept variance and MSE with respect to the center of each concept is in Fig. E.3 . The *kink* in gray lines is precisely the point where the model transitions from learning to represent the mean, to learning to explain the within-concept variance, clearly demonstrating two phases in learning: learning the right *scale* for the data (since initial model predictions may not match the true scale of data), thereby predicting the mean well, followed by learning the distribution of the data.

Fig. E.10 shows latent activation profiles for each concept and each SAE ($k = 32$ in TopK). Since TopK with $k = 32$ cannot allocate enough latents for large intrinsic dimension concepts, it increases activations on smaller number of concepts instead. Cosine similarities between SAE latent representations for pairs of data points, and pairs of latents across all datapoints, is shown for varying levels of sparsity in Fig. E.11, E.12 respectively. All SAEs (except JumpReLU) do a decent job at reducing correlations between pairs of data points, but in the latent correlation plots, we see how TopK fails to adaptively allocate latents to heterogenous concepts, especially at moderate levels of sparsity, while the other SAEs do well- have different sized blocks in block-structured matrix.

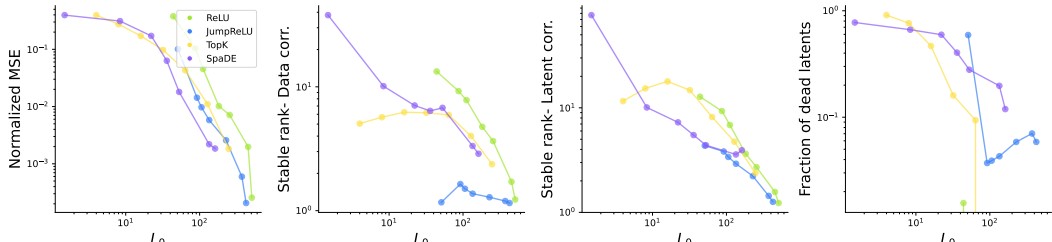

Figure E.7: Normalized MSE (normalized with variance of data), Stable ranks (of data correlations, latent correlations matrices), and fraction of dead latents as a function of sparsity ($L_0$) for the *heterogeneity experiment* (Sec. 5.2).

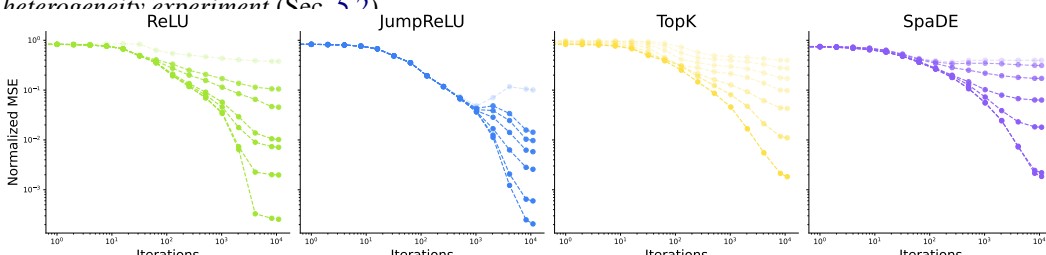

Figure E.8: Evolution of normalized MSE with training iterations for various SAEs on the *heterogeneity experiment*. Color intensity is proportional to $L_0$ (darker colors imply more dense SAE latents).

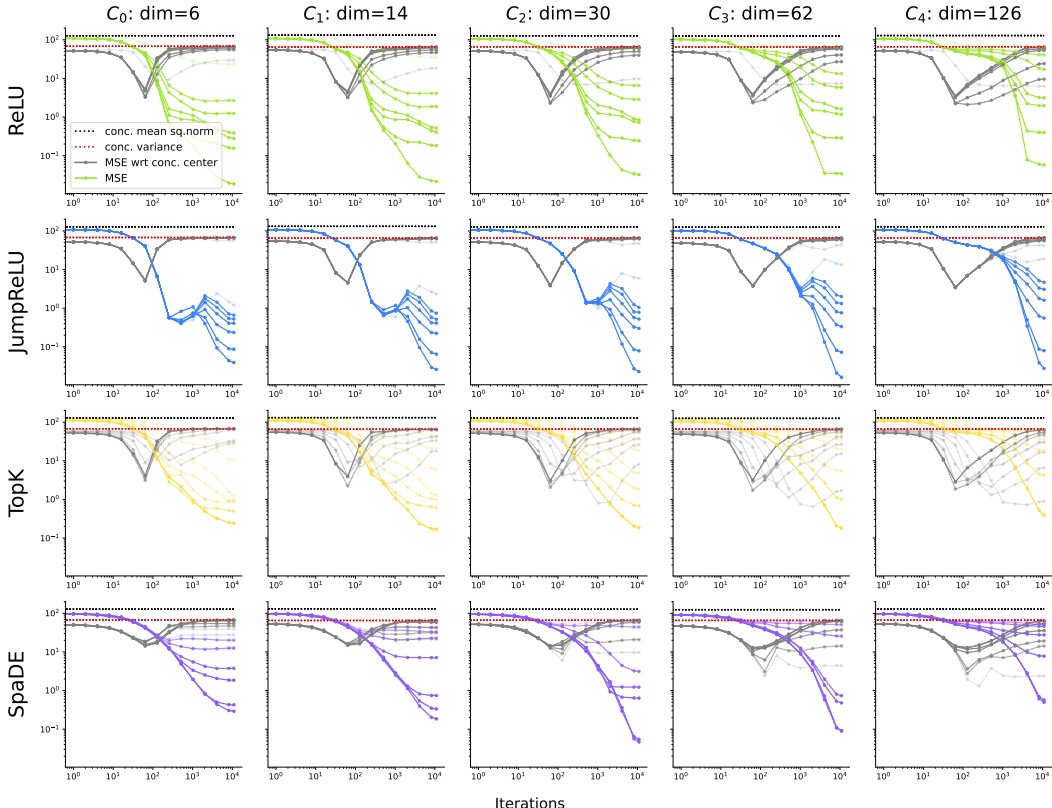

Figure E.9: Training dynamics for each concept (column) across SAEs (rows) for *heterogeneity experiment*: colored solid lines are MSE, with intensity of color proportional to $L_0$. Gray lines show MSE of SAE predictions with respect to the center of each cluster; intensity is again proportional to $L_0$. . Black dotted line shows the mean squared norm of each cluster, which would equal the MSE if the SAE predicted the origin for all datapoints. Red dotted line shows variance of each cluster, which again equals MSE if an SAE predicts the center of the cluster. Note that when a model reconstructs data well, MSE wrt cluster center equals the variance of the cluster (as observed here)

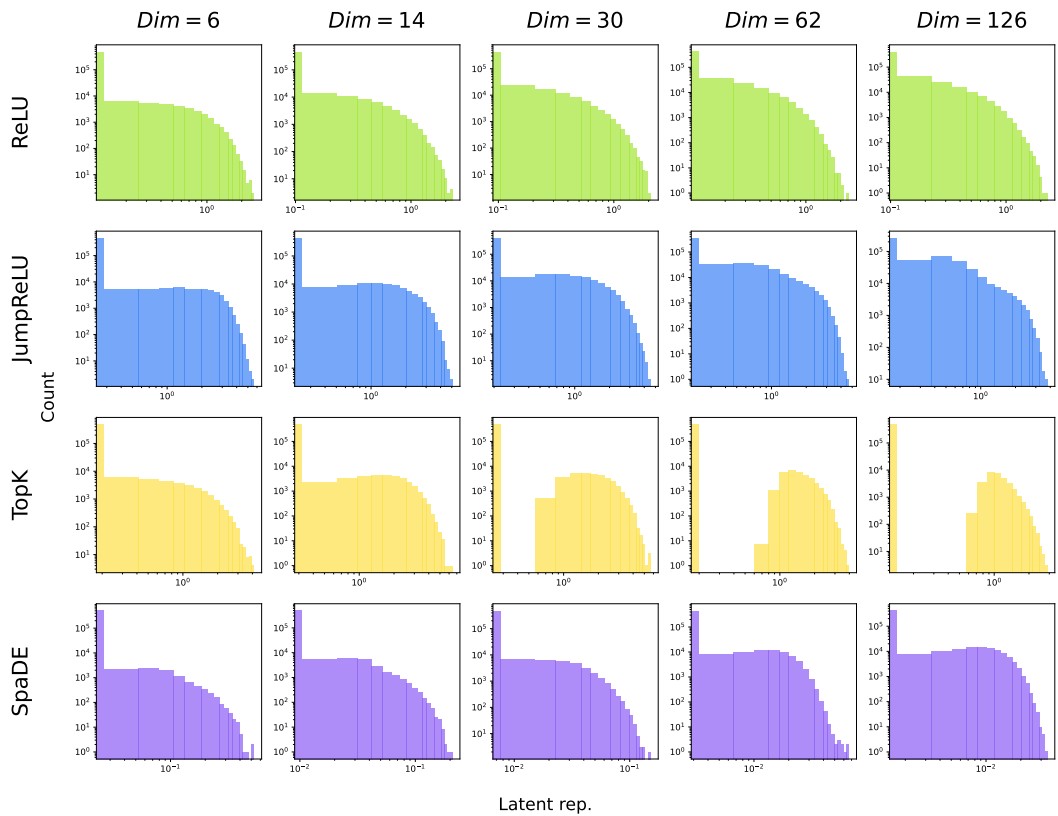

Figure E.10: Histogram of latent representations for each concept of various SAEs on the *heterogeneity experiment*.

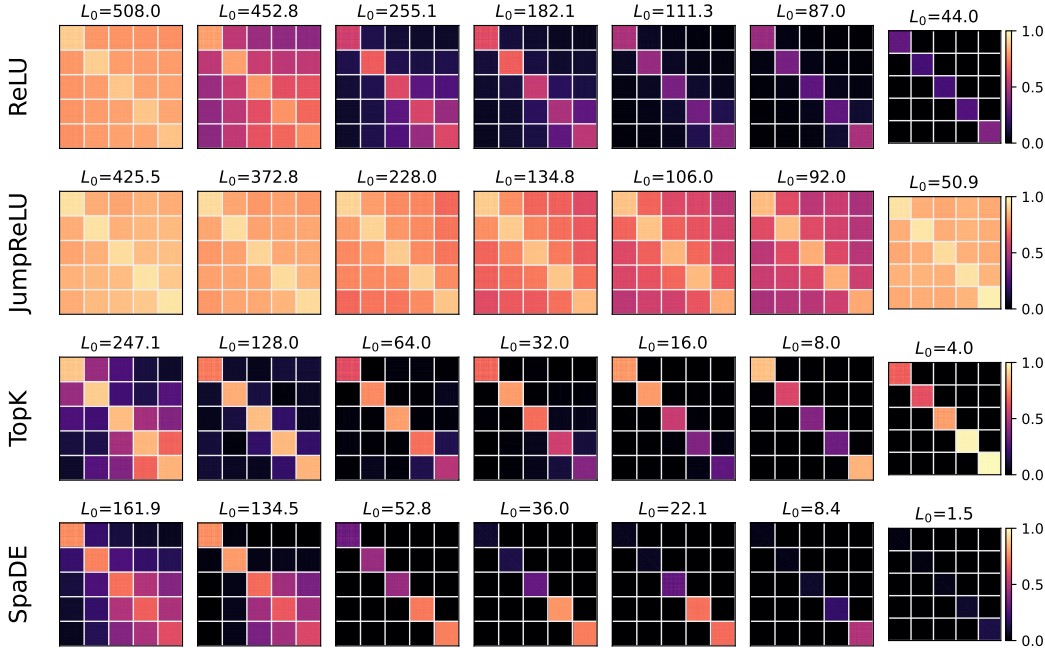

Figure E.11: Data correlations for various sparsity levels on the *heterogeneity experiment*: Pairwise cosine similarities between SAE latent representations of datapoints. White lines separate different concepts.

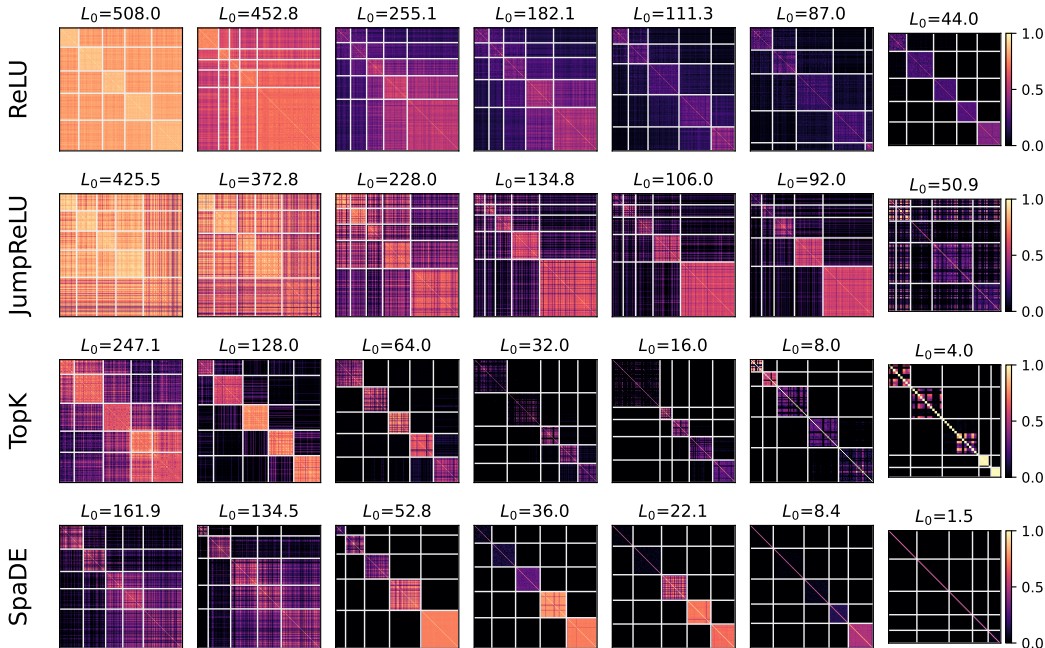

Figure E.12: Latent correlations for various sparsity levels on the *heterogeneity experiment*: Pairwise cosine similarities: pairwise cosine similarities between different SAE latents, computed across data from all concepts.

### E.3 Formal Language Experiments

In this section, we report several more results in the formal language experimental setup. Specifically, we show how with changing sparsity of the latent code, fidelity metrics, e.g., normalized MSE scales changes and stable rank of both data and latent correlations changes (Fig. E.13); how monosemanticity changes, i.e., how F1 scores averaged across latents and the concept they achieve maximum F1 score on change (indicating their specialization to that concept) (Fig. E.14 Left); and how percentage of dead latents change (Fig. E.14 Right). These results are repeated at the concept-level, i.e., at the level of parts-of-speech, in Figs. E.15, E.16. Inline with results on heatmaps demonstrating correlation between sparse codes of samples from different concepts and between vector denoting which samples a given latent activates for, we retrieve results in Fig. E.17, E.18. The results above are perfectly inline with our findings from the main paper, e.g., that SpaDE achieves highly monosemantic features. The new and intriguing results involve demonstrations of how effective SpaDE can be at discerning position of a concept (part-of-speech) in a sentence, when compared to other protocols which learn a more uniform representation.

Further, we also provide 2D and 3D PCA visualizations of different SAEs' retrieved latents in two different manners: (i) assess which datapoints a latent activates for and project it into a low-dimensional space identified using PCA, and (ii) assess which latents a datapoint activates, and project this activation vector. The former helps assess how monosemantic latents are, i.e., whether they activate for specific concepts, and the latter helps assess how specific latents are, i.e., whether a datapoint only activates a specific latent and hence there is no regularity present. Results show most SAEs, when they perform well, organize latents in a very structured manner (like a tetrahedron), but SpaDE succeeds at this throughout.

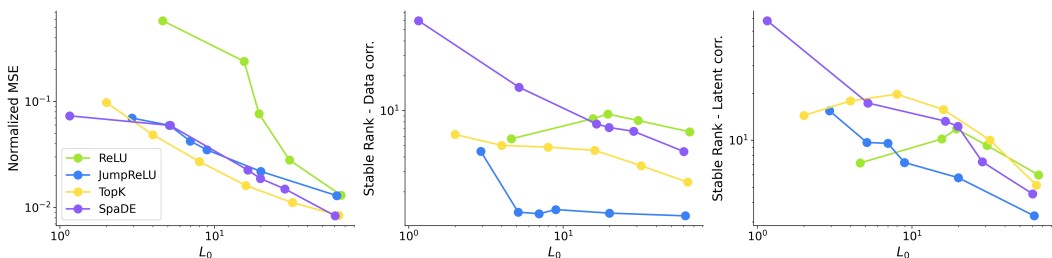

Figure E.13: Normalized MSE and Stable ranks as a function of sparsity in the Formal Language setup.

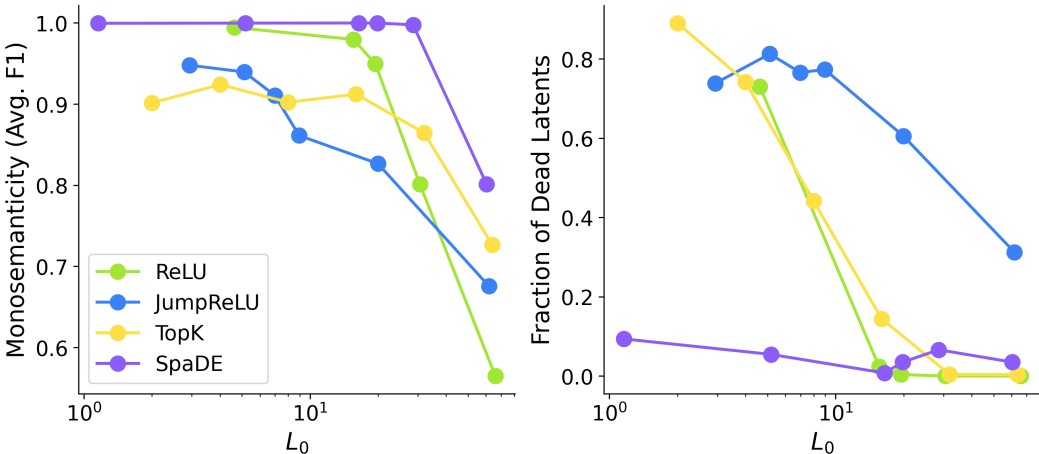

Figure E.14: Monosemanticity (F1 scores averaged over latents) and fraction of dead latents as a function of sparsity for different SAEs in the Formal Language setup.

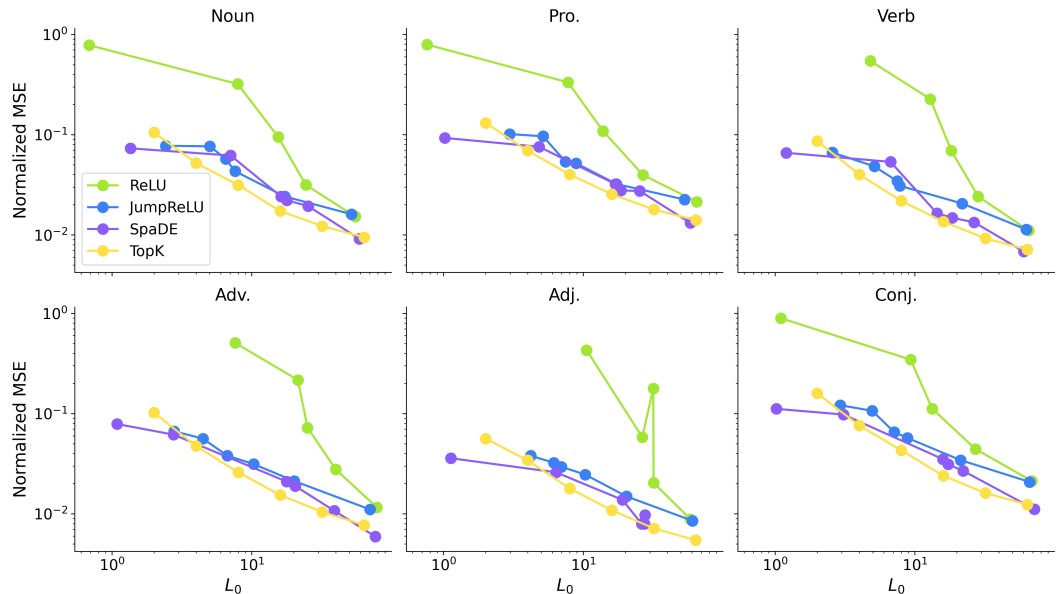

Figure E.15: Normalized MSE decomposed by concepts (parts-of-speech) and plotted as a function of sparsity in the Formal Language setup.

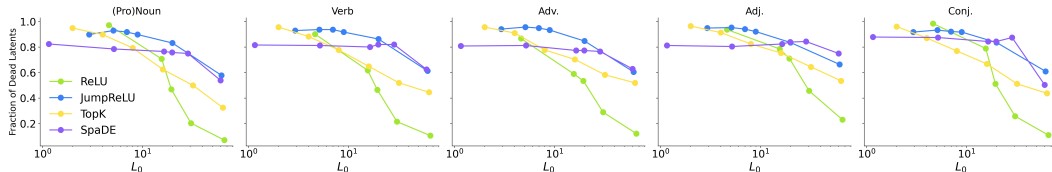

Figure E.16: Percentage of Dead Latents decomposed by concepts (parts-of-speech) and plotted as a function of sparsity in the Formal Language setup. Note that in such a concept-conditioned count of dead latents, one ends up counting both the latents that are always inactive and ones that are inactive for the specific concept under consideration.

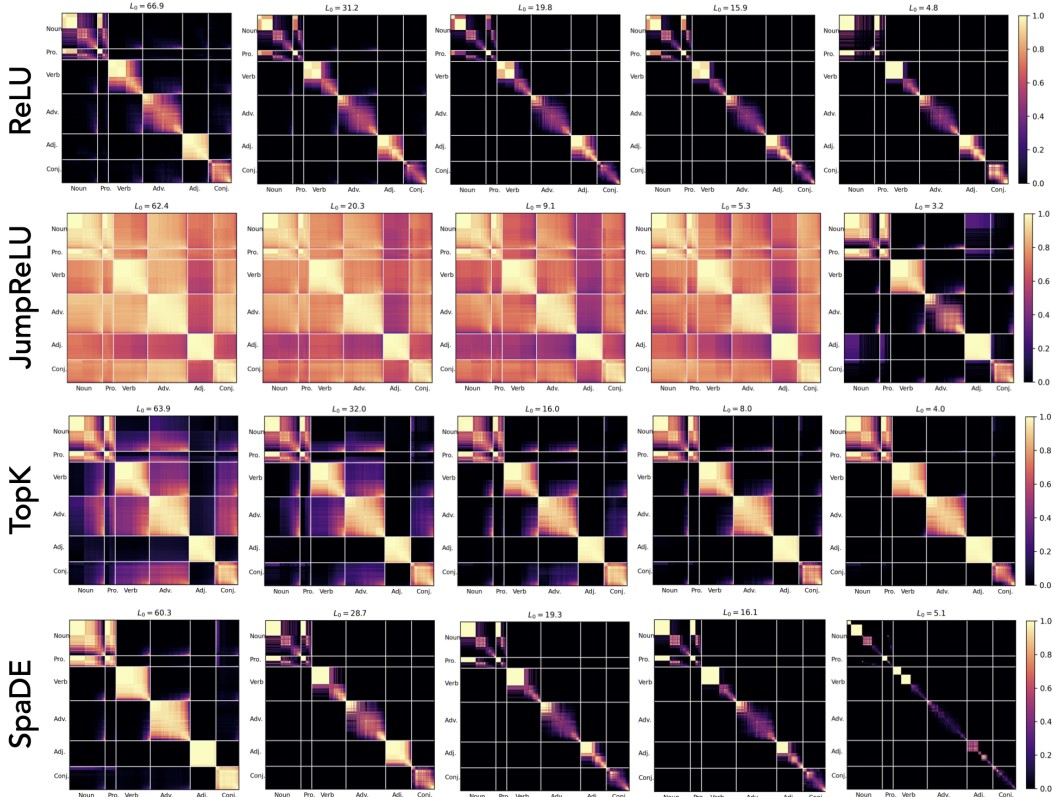

Figure E.17: Correlation between sparse codes of different concepts (parts-of-speech) in the Formal Language setup. Datapoints for different concepts are sorted according to which concept they come from (using a predefined order on the parts-of-speech) and according to their position in a sentence, hence highlighting position dependence. Lines demarcate boundaries at which tokens corresponding to different concepts start / end.

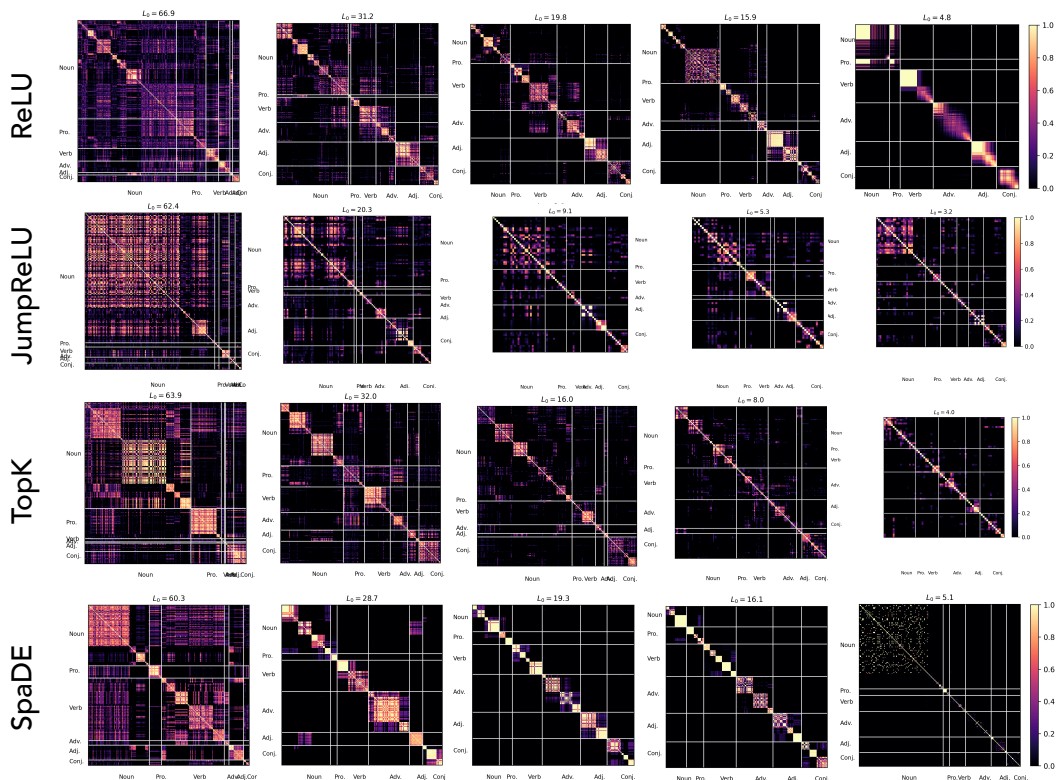

Figure E.18: Correlation between which datapoints a latent activates for in the Formal Language setup. Latents are sorted according to which concept (part-of-speech) they most strongly activated for (as measured using F1-score). White lines demarcate boundaries at which latents of different concepts start / end.

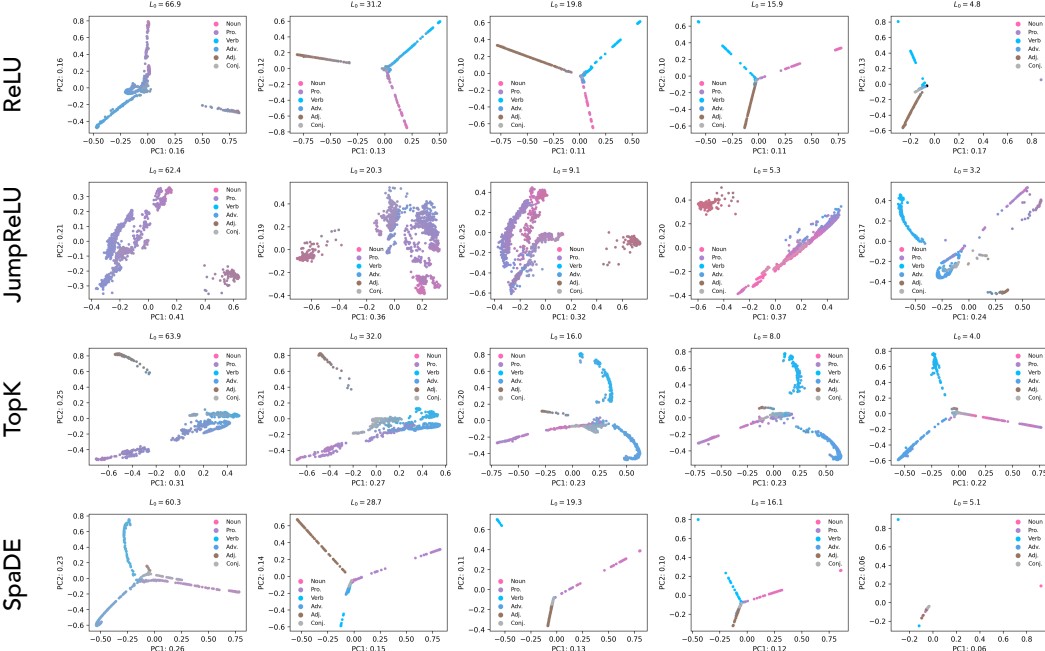

Figure E.19: 2D PCA visualization of sparse codes corresponding to different concepts (parts-of-speech).

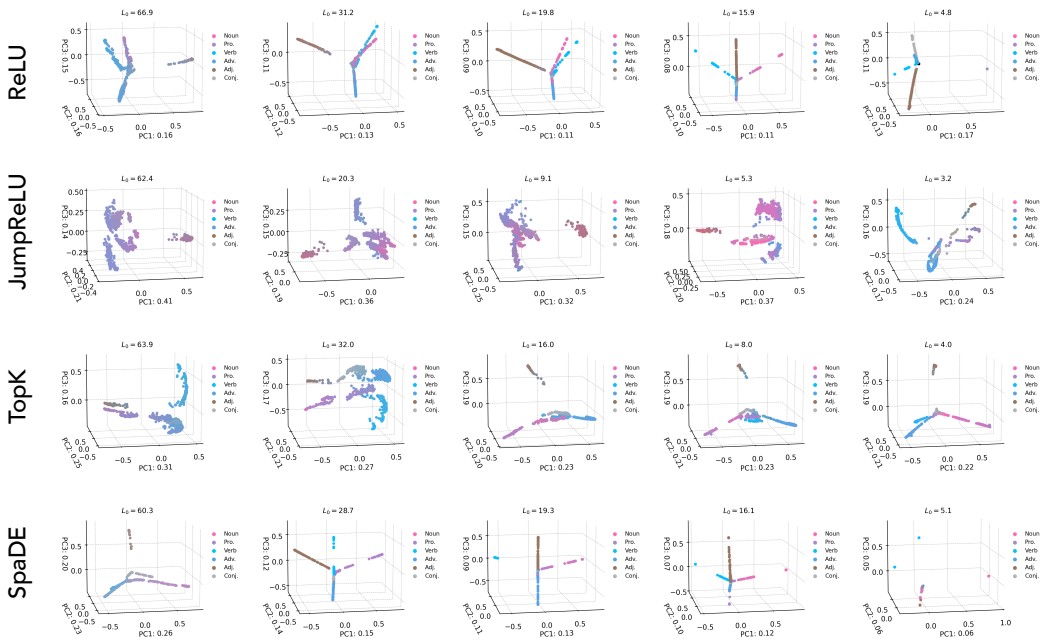

Figure E.20: 3D PCA visualization of sparse codes corresponding to different concepts (parts-of-speech).

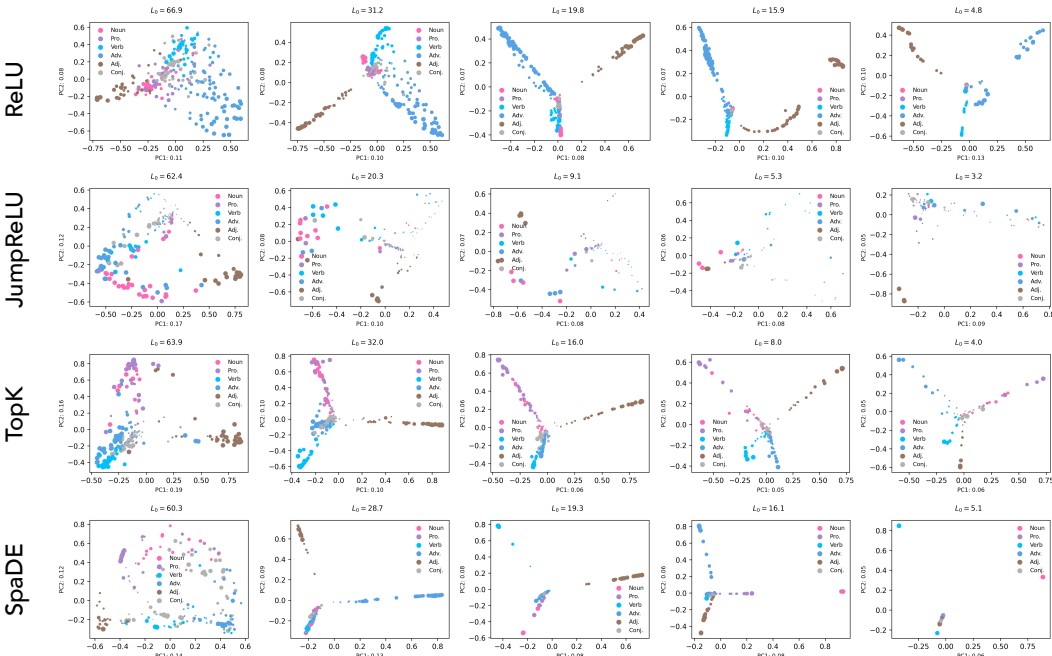

Figure E.21: 2D PCA visualization of a matrix whose elements capture which tokens a latent activates for. That is, which concepts (parts-of-speech) the latent is specialized towards, if any.

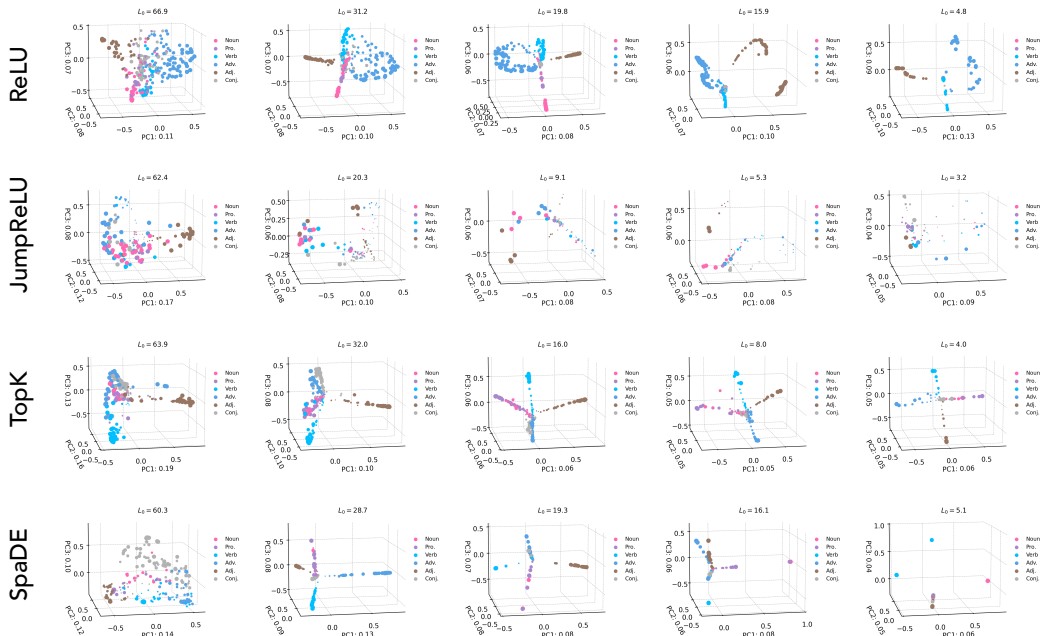

Figure E.22: 3D PCA visualization of a matrix whose elements capture which tokens a latent activates for. That is, which concepts (parts-of-speech) the latent is specialized towards, if any.

### E.4 Vision Experiment

In this section, we show, visually, the concepts SpaDE has learnt in the vision experiment, by visualizing feature attribution maps for inputs from each class from Imagenette. We perform this visualization for the top concepts for each class for five classes- Tench (Fig. E.23), Chainsaw (Fig. E.24), Church (Fig. E.25), Golf (Fig. E.26) and Springer (Fig. E.27)).

## Top-8 Concepts for **Tench**

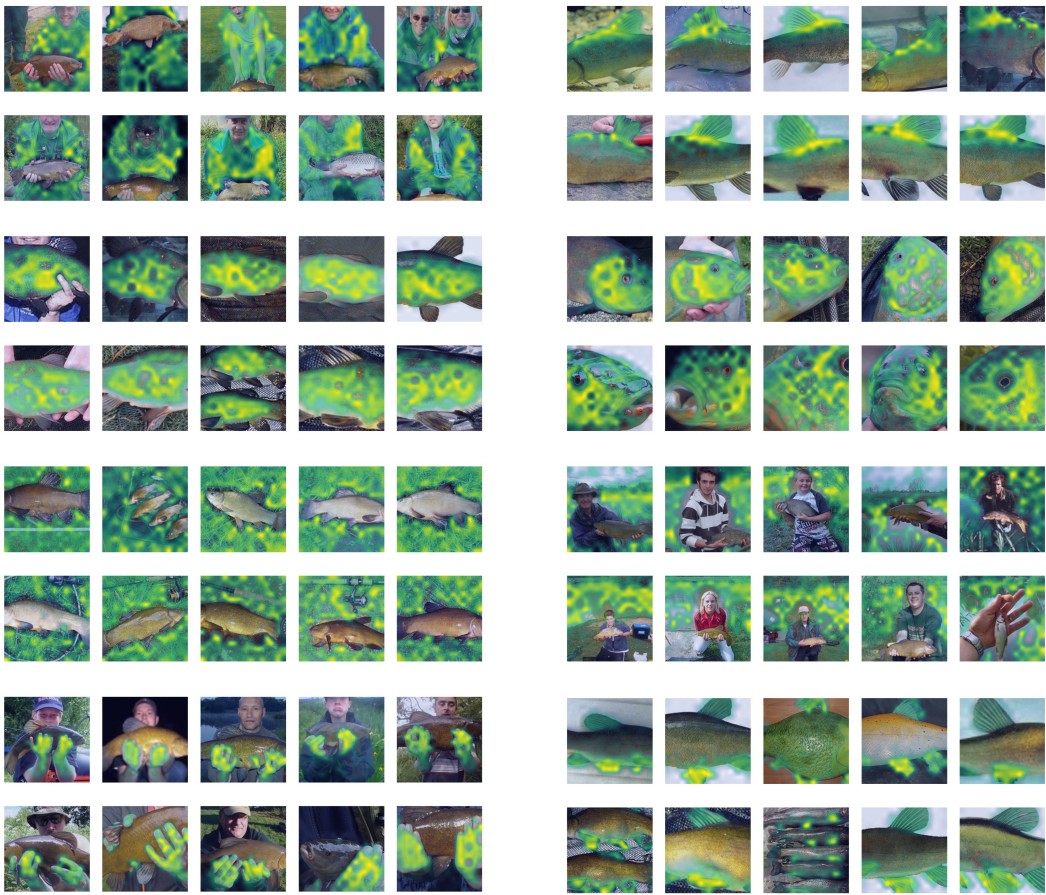

Figure E.23: Feature Attribution maps for monosemantic latents from SpaDE on the Tench class

## Top-8 Concepts for **Chainsaw**

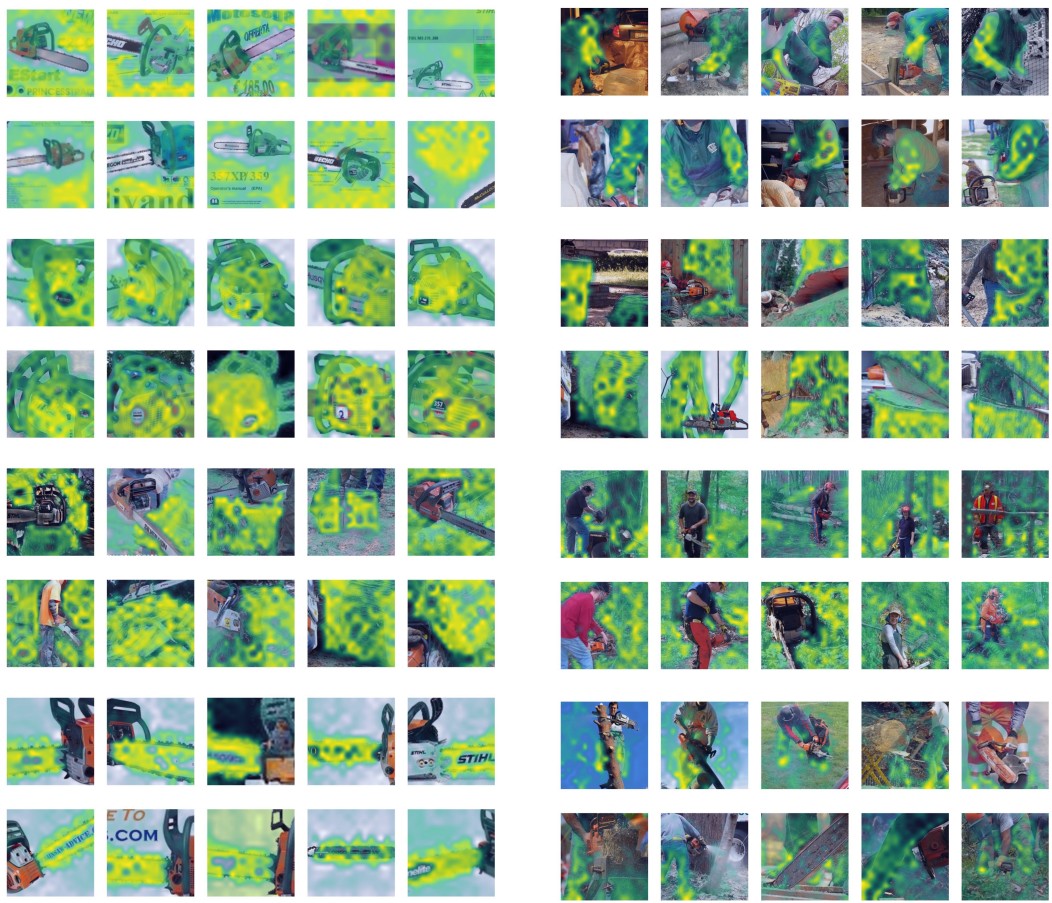

Figure E.24: Feature Attribution maps for monosemantic latents from SpaDE on the Chainsaw class

# Top-8 Concepts for **Church**

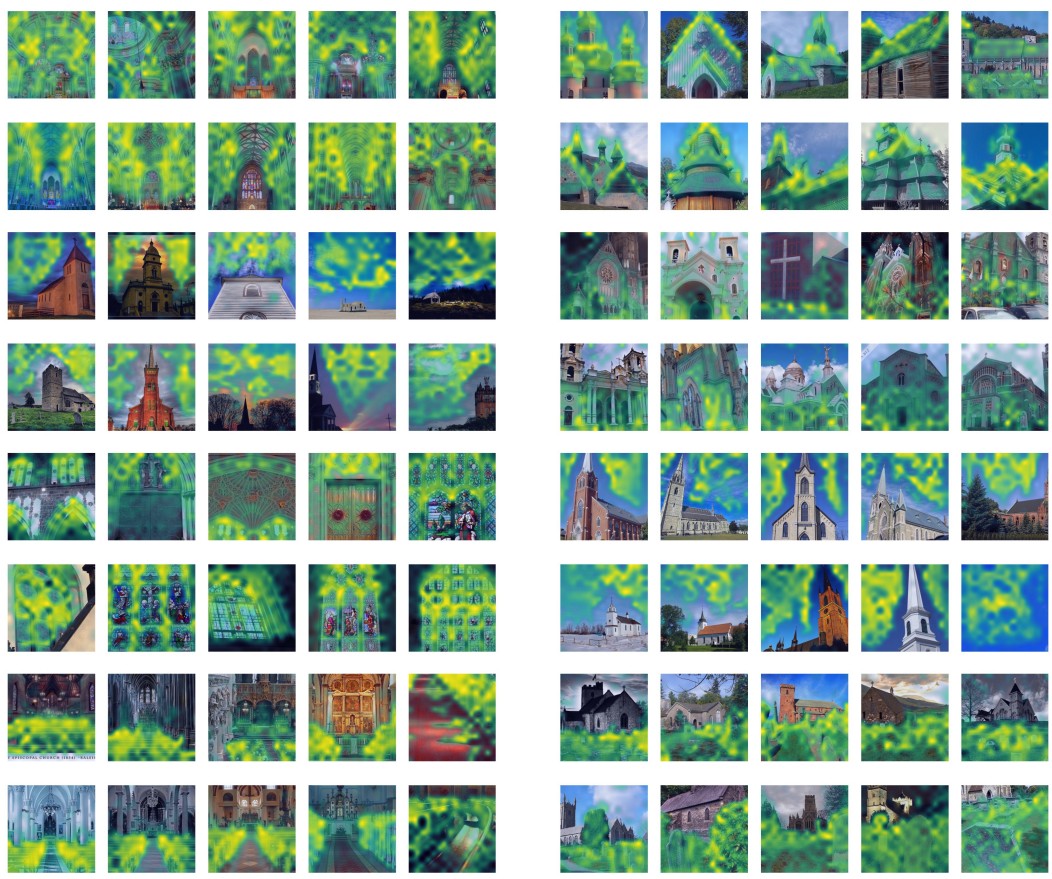

Figure E.25: Feature Attribution maps for monosemantic latents from SpaDE on the Church class

## Top-8 Concepts for **Golf ball**

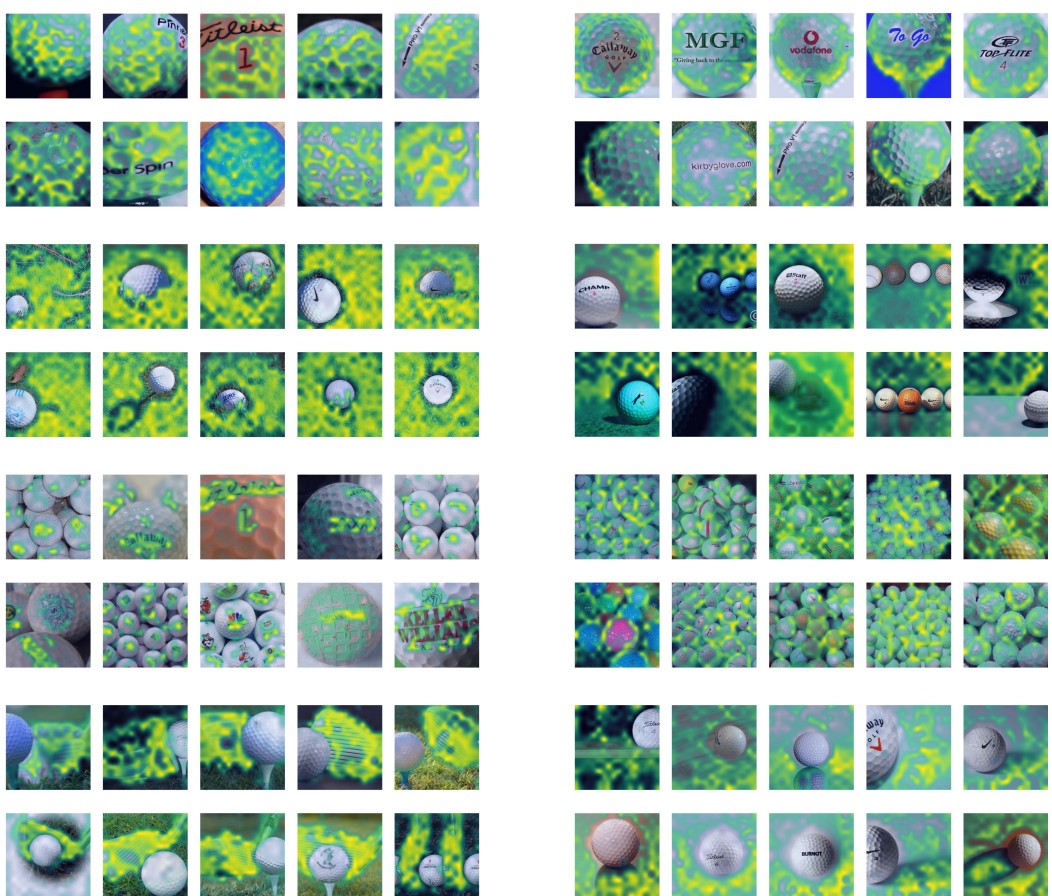

Figure E.26: Feature Attribution maps for monosemantic latents from SpaDE on the Golf class

# Top-8 Concepts for **English springer**

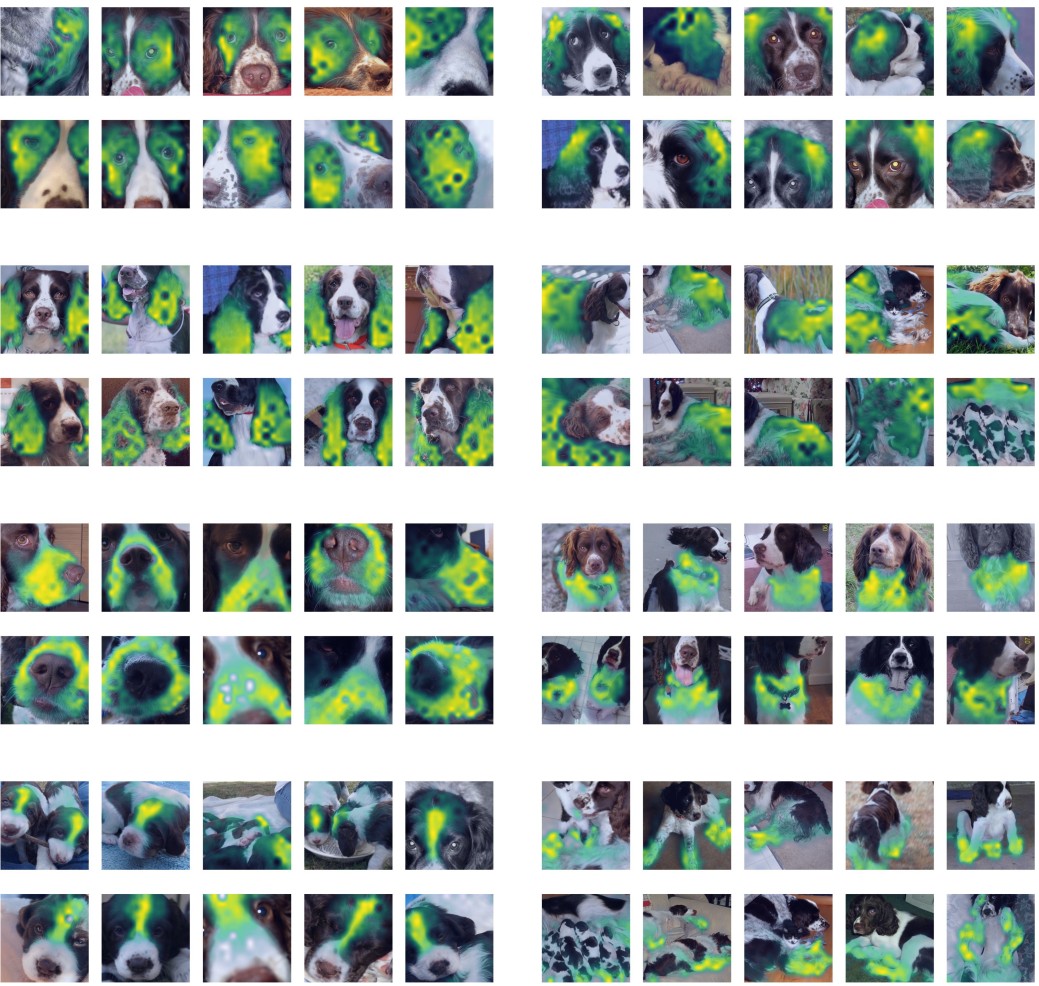

Figure E.27: Feature Attribution maps for monosemantic latents from SpaDE on the Springer class

