# OpenReview forum: "Projecting Assumptions: The Duality Between Sparse Autoencoders and Concept Geometry"
_NeurIPS.cc/2025/Conference — NeurIPS 2025 poster_

### Official Review · Reviewer_Bs68 · 2025-07-01

**Clarity:** 2
**Significance:** 3
**Originality:** 3
**Rating:** 4
**Confidence:** 3

**Summary:**

This paper critically examines the ability of Sparse Autoencoders (SAEs) to reveal meaningful concepts in neural network representations. While SAEs are commonly used in interpretability research, the authors argue that their ability to uncover concepts is fundamentally constrained by the structural assumptions built into their architectures. The paper frames SAE learning as a bilevel optimization problem, revealing how design choices, particularly the non-linearities used and the structure of the latent space, act as inductive biases that filter which types of concepts can be detected.

The authors identify two core challenges to concept discovery: (1) real-world concepts may not correspond to linear directions but may require modeling non-linear manifolds, and (2) concepts may vary in complexity and intrinsic dimensionality, i.e., heterogeneity. Through extensive experiments, ranging from controlled synthetic setups to semi-synthetic and natural datasets across both vision and language domains, the authors show that standard SAEs often fail to recover ground truth concepts when these properties are not aligned with their architectural assumptions. To address this, they propose a novel SAE variant that better accommodates heterogeneity and non-linearity, demonstrating improved concept recovery and supporting their theoretical claims.

Overall, the paper challenges the notion of a universal SAE architecture and emphasizes the importance of architecture-specific choices in concept-based model interpretability.

**Questions:**

1. On the necessity of non-linear concept modeling:
Prior work (e.g., [42]) suggests that many meaningful concepts are approximately linear. Could you clarify why it is essential to model non-linear concepts rather than decomposing them into combinations of simpler linear ones? Maybe the non-linear concept appears a linear one in a later layer?

2. On concept heterogeneity and latent usage:
You argue that complex, heterogeneous concepts activate multiple latent units. How does this square with the goal of achieving monosemanticity, where ideally each latent corresponds to a single interpretable concept? Do you view these complex concepts as superpositions of atomic ones?

3. Assumption of one concept per input:
Many of your experiments seem to rely on the assumption that each data point corresponds to a single underlying concept. Could you justify this assumption, or discuss how your approach generalizes to settings where multiple concepts are present simultaneously?

4. On interpretability and steerability goals:
Since the paper is motivated by the desire to improve interpretability and steerability, have you considered evaluating your method on benchmarks or tasks that directly measure these goals? Without such evaluation, how should we assess the practical gains?

5. On expectations of concept recoverability:
You use pre-defined, often class-based concepts to evaluate SAE performance. However, models like DINO are trained without labels and may not encode class concepts directly. How do you justify the use of these expected concepts, and do you have evidence that the model itself encodes them in a disentangled or monosemantic way?

6. Clarification on Experiment 1 setup and goals:
Could the non-linear clusters in Experiment 1 not be interpreted as mixtures of simpler linear Gaussians? If so, why is it preferable to directly recover the non-linear shapes, rather than their linear components?

**Ethical Concerns:**

["NO or VERY MINOR ethics concerns only"]

**Final Justification:**

This paper delivers valuable theoretical and empirical analysis on how architectural assumptions in Sparse Autoencoders constrain the kinds of concepts they can recover, offering a constructive new variant (SpaDE) as an example. The work is notable for its breadth of experiments, spanning vision and language tasks. However, the paper is dense, and at times the motivation for individual experiments and the interpretation of results are not sufficiently clear, leaving readers uncertain about what an SAE should ideally recover in a given scenario (e.g., whether a toy cluster should represent a pure monosemantic concept or a composite of atomic ones, and when non-linear concepts should be expected). These ambiguities partly reflect open questions in sparse dictionary learning and concept interpretability, but they also make it harder to assess the practical impact of the findings. The authors’ focus is on improving theoretical understanding rather than benchmarking interpretability or steerability gains in real-world large-scale settings, which is a valid and important aim, though its downstream utility remains to be demonstrated. Overall, the paper makes a meaningful conceptual contribution to the interpretability literature, but could benefit from clearer framing of expectations, stronger guidance to the reader through the dense material, and discussion of how the improved understanding might concretely inform the design and evaluation of SAEs in practice.

**Limitations:**

Limitations are discussed, but no negative societal impacts are mentioned.

**Paper Formatting Concerns:**

No concerns.

**Quality:**

3

**Strengths And Weaknesses:**

Strengths:
- The paper presents a broad and well-designed experimental evaluation, spanning both vision and language domains, and progressing from controlled toy tasks to complex real-world settings.
- The presentation quality is excellent, with clear visualizations and intuitive illustrations that make complex ideas accessible.
- Theoretical insights are not only discussed abstractly but also used constructively to develop a new SAE architecture that performs better empirically.
- The paper brings valuable theoretical perspective by analyzing how architectural choices in SAEs induce biases over feature geometry, including latent component receptive fields and separability assumptions. This kind of reflection is important for developing more rigorous interpretability methods.

Weaknesses:
- It is unclear whether non-linearity and heterogeneity are fundamentally problematic for existing SAEs. Prior work [42] suggests that many meaningful concepts are in fact linear, and standard SAEs are already capable of modeling multi-dimensional representations when needed. In many interpretability settings, linear concepts are preferable due to their simplicity.
- The authors argue that complex concepts must activate many latent units, which seems to contradict the common goal of having one latent per concept. This only makes sense if such “complex” concepts are composed of more atomic parts, then each latent might represent one of these atomic concepts.
- Many experiments assume that each data point reflects a single concept. This is an unrealistic assumption, as data often contains multiple overlapping features. This weakens the motivation for their SAE architecture, the overall conclusion, and practical significance of the evaluation.
- In Experiment 1, the “non-linear” Gaussian clusters could reasonably be interpreted as combinations of the “linear” Gaussians. The authors argue that only by discovering the "non-linear" clusters one achieves monosemanticity, but this is debatable. A simpler decomposition might actually be preferable.
- The goal of SAEs is to improve interpretability and steerability, but the paper does not evaluate either in terms of established benchmarks or automated interpretability measures. As a result, it’s unclear how the proposed model advances the ultimate goals of this line of work. Or how each SAE architecture is limited or improves interpretability/steerability.
- The evaluation focuses on recovering pre-defined concepts, but it is questionable whether these are meaningful or expected (e.g., whether ImageNet classes are the right targets for Dino representations). Arguably, the true latent features may be more abstract.
- Some theoretical claims and explanations lack clarity or are not well-motivated. For example, the reason why the TopK selection yields angular separation could be explained more directly.


Minor Comments:
- In Eq. (1), the variable z depends on the input x, but this dependency is not explicitly shown.
- In Eq. (2), consider noting that the TopK SAE (used in the experiments) does not include an encoder bias term.
- Table 1 reuses the variable x for the latent code, which previously denoted the input. This is confusing.
- Line 87: Is “orthogonal” projection really orthogonal? Please check.
- Definition 3.2: The term “neuron” is ambiguous since the paper focuses on latents, not neurons. Also, $f^{(k)}$ could be interpreted as a derivative. Is it $\mathbb{R}^d \rightarrow \mathbb{R}$ or otherwise?
- Line 127: Typo: should be “is” instead of “in”.
- Line 133: The term “onion” feature is unclear, consider explaining or rephrasing.
- Table 3: Could TopK SAE not handle some heterogeneity by selectively activating k latents? Never more than $k$ latens can be active, but fewer could be.
- Potentially use formal equation references with parentheses, e.g., “Eq. (1)” rather than “Eq. 1”.
- Figure 5 caption: Missing whitespace at the end of a sentence.
- Line 235: What does PCFG stand for? Please define.
- Inconsistent casing: “Row” is capitalized in Sec. 5.3, but lowercase in Sec. 5.4.
- Line 245: Capitalize “MSE”?
- Figure 6: In row (b), TopK and SpaDE appear visually very similar (the sparsity-MSE trade-off). This similarity could be addressed or acknowledged.

---

> ### Author Rebuttal · Authors · 2025-07-31
>
> We thank the reviewer for their detailed and insightful review and comments! We greatly appreciate that they find our work provides “valuable theoretical perspective”, “presentation quality is excellent”, “broad and well-designed experimental evaluation”. We also appreciate that they highlighted our main message: SAEs’ ability to extract concepts is constrained by structural assumptions in their architecture.
>
> We address specific comments and questions below.
>
> > **unclear whether non-linearity and heterogeneity are fundamentally problematic for existing SAEs.**
>
> Thank you for the comment! We break down our response into two parts, addressing the comments on nonlinearity and heterogeneity one-by-one.
>
> - **Non-linearity**: While the reviewer is correct that many “meaningful concepts” encoded in neural networks have been shown to be linear, i.e., they satisfy the linear representation hypothesis (LRH), there is growing evidence for models encoding concepts outside the scope of LRH (e.g., magnitude-dependent onion features, Csordas et al 2024). In fact, we emphasize that no general proofs force a neural network to represent all concepts in a linearly accessible manner. To this end, we both formally and empirically show that ReLU and JumpReLU SAEs (though not TopK SAEs), which assume linear separability of concepts, fail at recovering nonlinearly separable concepts!
>
>
> - **Heterogeneity**: The reviewer noted that “standard SAEs are already capable of modeling multi-dimensional representations when needed”; however, we emphasize that this statement is only true for ReLU and JumpReLU SAEs! We show in our work that TopK SAEs fail to uncover low-dimensional concepts from a dataset which contains concepts with intrinsic dimensionality lower than the value of K in the topK activation function (Figure 6a).
>
> More broadly, we emphasize the core contribution of our work: when neural networks encode concepts outside the scope of LRH, an SAE’s success at extracting them will depend on assumptions it makes about concept structure!
>
>
> > **Complex, heterogeneous concepts activate multiple latent units: at odds with monosemanticity?**
>
>
> We would first like to clarify that we do not use the term “complex concepts” in our work. We believe the reviewer is referring to our analysis of multi-dimensional concepts—which are often composed of several atomic (1D) concepts. Collapsing these into a single latent risks losing access to meaningful subcomponents. For instance, Pan et al. (2025) show that “safety” involves dimensions like “refusal behavior,” “hypothetical narrative,” and “role-playing,” which are also useful beyond safety (e.g., persona control). Thus, it is natural for such concepts to occupy a multi-dimensional subspace rather than a single direction.
>
> > **Experiments assume each data point has a single concept: justify this assumption. Does your approach generalize to multiple concepts being present simultaneously?**
>
>
> Thank you for the question. Our claim about SAEs being biased towards certain concepts is a property of their architecture, and thus holds for *any* data distribution. We use non-overlapping concepts as a simple case with real-world examples (such as mutually exclusive classes or children of the same parent in a hierarchical generative process) to highlight the limitations of existing SAEs. Our theoretical claims about SAE biases will also hold for overlapping concepts if we generalize the definition of separability from (current) *distinguishing one concept from other concepts (akin to multiclass classification)* to (improved) *distinguishing between the presence and absence of a concept (binary classification for each concept)*. We will formally highlight the significance of SAE biases in the presence of overlapping concepts in future work.
>
>
> > **Could the non-linear clusters in Experiment 1 not be interpreted as mixtures of simpler linear Gaussians? If so, why is it preferable to directly recover the non-linear shapes, rather than their linear components?**
>
>
> We apologize that this was not clear. To briefly clarify, in Experiment 1 we study a mixture of nonlinearly separable and linearly separable gaussian clusters, where each cluster is defined as its own concept. This is a toy abstraction of “days of the week” (Engels et al 2024), arranged on a two-dimensional circle in model activations. In this setting, SAEs are expected to recover each concept (cluster) in a monosemantic fashion, since the goal for SAEs is to develop interpretable decompositions of model representations, not necessarily simpler ones. Thus, it is at times necessary to directly recover “non-linear” concepts, such as the magnitude-dependent “onion-features” from Csordas et al 2024, instead of decomposing them into simpler “linear” concepts.
>
> More broadly, concepts are defined by human interpretation and knowledge on data, independent of their geometric structure in model activations.
>
> > **Since the paper is motivated by the desire to improve interpretability and steerability, have you considered evaluating your method on benchmarks or tasks that directly measure these goals? Without such evaluation, how should we assess the practical gains?**
>
> Thank you for raising this important point. Our main motivation in this work is to highlight architectural limitations of existing SAEs in achieving monosemanticity, and not to improve interpretability and steerability through new SAE architectures. SpaDE serves as a constructive example—designed to incorporate properties like nonlinear separability and concept heterogeneity—and *not* as a superior SAE. Our work aims to improve understanding of SAEs as interpretability tools and differs fundamentally from efforts focused on benchmarking new architectures on large-scale LLMs.
>
> > **Models like DINO are trained without labels and may not encode class concepts directly. How do you justify the use of these expected concepts, and do you have evidence that the model itself encodes them in a disentangled or monosemantic way?**
>
> We note the use of class-based concepts (using a subset of classes from ImageNet in Section 5.4) is justified since prior work [1] shows that such concepts are disentangled and hence linearly recoverable from DINOv2’s representations. That work in fact benchmarks different SAEs’ ability to extract these representations in order to compare them! This is the reason why we use pre-defined concepts as well.
>
> [1] Fel, Thomas, et al. "Archetypal sae: Adaptive and stable dictionary learning for concept extraction in large vision models." arXiv preprint arXiv:2502.12892 (2025).
>
>
> > **The reason why the TopK selection yields angular separation could be explained more directly.**
>
> Thank you for highlighting this point! We will include more intuitive explanations for the assumptions of SAEs and clarify the theory in the final version of the paper. The “angular separability” assumption of TopK SAE comes from the derived receptive field (App. D.2.2, line 867).
> For latent m in TopK SAE to fire, its linear transform in encoder must exceed at least $D-K$ other linear transforms (where D is the SAE width). This gives us an intersection of half-spaces through the origin, leading to angular separation.
>
> ---
> ---
>
>
> **Minor comments: typos and style inconsistencies**
>
> Thank you so much for highlighting these typos and style inconsistencies! We will incorporate all necessary changes in the final version of our paper.
>
>
> > **Figure 6: In row (b), TopK and SpaDE appear visually very similar (the sparsity-MSE trade-off).**
>
> Good observation! While the similar sparsity–MSE tradeoff exhibited by TopK and SpaDE in Figure 6 is interesting, the correlation maps, which show more interpretable structures emerging in SpaDE despite its similar reconstruction abilities as TopK validate our claims better.
>
> > **Line 235: What does PCFG stand for? Please define.**
>
> We apologize this was unclear! PCFG stands for “Probabilistic Context-Free Grammars”, a formal model of language often used to study its syntactic properties. See Appendix C.3 for further details.
>
>
> > **Line 87: Is “orthogonal” projection really orthogonal?**
>
> Yes! An orthogonal projection of a vector onto a set S is defined in Eq. (3). This is a generalization of the idea of orthogonal projection onto a line / plane (see Axler, Linear Algebra Done Right, Chapter 6, Section 6C) to arbitrary sets S (which are closed).
>
> >**Definition 3.2: The term “neuron” is ambiguous since the paper focuses on latents, not neurons.**
>
> Apologies for the confusion! In Definition 3.2, we borrow the terminology “neurons” from neuroscience, which is the original motivation for receptive fields (line 108, 109). In the context of our work, we refer to “neurons” of the SAE encoder.
>
>
> > **Line 133: The term “onion” feature is unclear.**
>
> We note our use of the term “onion features” is inline with the paper that first coined the term (Csordas et al, 2024). In that work, the authors show RNNs learn to encode input concepts as the magnitude of a feature—not its direction, akin to peels of an “onion”.
>
> > **Table 3: Could TopK SAE not handle some heterogeneity by selectively activating k latents?**
>
> Good question! In principle, we agree it is *possible* that TopK SAEs learn to activate less than K latents for reconstruction. However, as seen in Figure 6(a), TopK does not adapt its sparsity level (y-axis) to match the dimension of the concepts (x-axis), while it may be able to demonstrate low MSE (mean squared error, color coded) on these concepts.
>
> **Summary**: We thank the reviewer again for their insightful questions, which led us to further clarify our main claims, describe the motivation behind experimental setups better, and justify how our experiments support claims! We hope our response has sufficiently addressed these points, and if so, we hope this encourages the reviewer to increase their score! We would be happy to address any further questions in the discussion period.

---

> > ### Comment · Reviewer_Bs68 · 2025-08-07
> >
> > Thank you to the authors for their detailed and thoughtful feedback. I appreciate the clarifications provided.
> >
> > One point that would benefit from further refinement is clarifying when the goal is to capture compositions of atomic (1D) concepts versus when the focus is on discovering inherently multi-dimensional concepts. Making this distinction more explicit would help contextualize the method and its intended applications.
> >
> > I am hopeful that this paper will inspire further research on the theoretical and geometrical aspects of sparse dictionary learning, which remains an underexplored but promising direction in interpretability.
> >
> > That said, I remain somewhat unconvinced about the practical usefulness of the method as it stands, or whether it directly addresses pressing real-world interpretability challenges. A more in-depth discussion of current limitations, particularly around the problem of the LRH, real world applicability would strengthen the paper significantly.
> >
> > If such a discussion and clarifications were added, I would be happy to raise my score by one point.

---

> ### Author Response · Authors · 2025-08-07
>
> We thank the reviewer for their continued engagement---we are glad to see our clarifications were helpful! We are excited to see the reviewer's comment that "I am hopeful that this paper will inspire further research on the theoretical and geometrical aspects of sparse dictionary learning, which remains an underexplored but promising direction in interpretability"
>
> We respond to additional comments next.
>
> > One point that would benefit from further refinement is clarifying when the goal is to capture compositions of atomic (1D) concepts versus when the focus is on discovering inherently multi-dimensional concepts.
>
> We certainly agree and realized based on reviewer comments that the scope of our arguments (atomic concepts / multi-dimensional ones) could be better emphasized. To this end, we will rework Section 5 by adding addition introductory paragraphs that clarify the scope of the experiments discussed therein, relating back to the terminology and examples introduced in Section 4 (Separability and Heterogeneity).
>
> > That said, I remain somewhat unconvinced about the practical usefulness of the method as it stands, or whether it directly addresses pressing real-world interpretability challenges. A more in-depth discussion of current limitations, particularly around the problem of the LRH, real world applicability would strengthen the paper significantly.
>
> Certainly! While we did include a discussion in Section 6 (Discussion and Limitations) that describes how we are not arguing for SpaDE to be a "better" SAE, i.e., one that can be used for explaining any model behavior, we would love to bring some of that discussion back into the main text. To this end, we will use the extra page allowed for camera-ready version to state explicitly when introducing SpaDE that we propose it specifically to validate our claims, i.e., to demonstrate the duality between data assumptions and SAE architecture. SpaDE is a dual for nonlinear separability and heterogeneity, and hence should outperform other SAEs when these geometrical constraints are satisfied, which, based on recent work, often are (we will add further examples to make this point clearer). Meanwhile, when these constraints are not satisfied, we can expect other SAEs to perform on par or at times outperform SpaDE. Even then, however, each SAE has its own implicit assumptions and that will be reflected in the performance comparisons (e.g., the fixed sparsity assumption of TopK can affect its ability to handle mutlidimensional concepts).
>
> ----
>
> **Summary:** We hope our planned edits help address the reviewer's concerns. Please let us know if there are any further questions!

---

### Official Review · Reviewer_E5bk · 2025-07-03

**Clarity:** 4
**Significance:** 3
**Originality:** 4
**Rating:** 5
**Confidence:** 4

**Summary:**

The authors show that different SAE architectures implicitly bake in different assumptions about the structure of model representations, via an analysis of the SAE training objective. They then develop a new architecture, called SpaDE, which enables concepts to be discovered even when they exhibit nonlinear separability and heterogeneity. In various settings, the authors find that SpaDE can improve over existing SAE architectures.

**Questions:**

* This paper focuses on SAEs, but there is some momentum in mechanistic interpretability away from SAEs on their own and more towards models which capture aspects of model computation like CLTs. Do you have any thoughts on the applicability of your framework to more general types of interpreter models? It seems like the analysis could apply pretty directly, since CLT encoders are the same as SAE encoders. But more broadly, I wonder if there are other more general ways that your point -- that interpreter model architecture determines what we can discover about models -- could apply to the design of other sorts of sparse interpreter models.
* Do you think your work could explain any potentially-pathological results that have been described about SAEs? Could our choice of architecture be leading to feature splitting, or to non-atomic features being learned, that could be ameliorated with a different choice of architecture?

**Ethical Concerns:**

["NO or VERY MINOR ethics concerns only"]

**Final Justification:**

I found the paper to be solid from the beginning, and the authors' response was mostly just answering some higher-level questions I had had. I'm keeping my score at a 5. The paper is well-executed and interesting.

**Limitations:**

yes

**Quality:**

4

**Strengths And Weaknesses:**

### Strengths
* The paper is exceptionally well-written and well-constructed, with beautiful figures and diagrams.
* The core point, that SAE architectures make implicit assumptions about the data, is a highly valuable one. The paper provides a useful framework for reasoning about the assumptions that we make when designing interpreter models. The SAE architecture that its analysis leads to is also an interesting one.

### Weaknesses
* The language model evaluations of SpaDE are a little lighter than what most papers introducing an SAE architecture conduct (scaling laws, sparsity vs. reconstruction frontiers, etc.). It would be good to ultimately have a more detailed empirical study of SpaDE on full-scale LLMs, but I recognize that SpaDE is not the only contribution of the work, and the broader theoretical/conceptual point about the relationship between SAE architecture and assumptions about the data is arguably more important.

---

> ### Author Rebuttal · Authors · 2025-07-30
>
> We thank the reviewer for their review and feedback! We are glad that they found our work’s “core point … is highly valuable”, and the paper “exceptionally well-written”, with “beautiful figures”. We appreciate that they highlighted our core contributions: 1) different SAEs make different assumptions about concept structure,  and 2) SpaDE, a new SAE, captures concepts satisfying nonlinear separability and heterogeneity.
>
> We provide responses to specific questions and comments below.
>
>
> > **The language model evaluations of SpaDE are a little lighter than what most papers introducing an SAE architecture conduct (scaling laws, sparsity vs. reconstruction frontiers, etc.). It would be good to ultimately have a more detailed empirical study of SpaDE on full-scale LLMs, but I recognize that SpaDE is not the only contribution of the work, and the broader theoretical/conceptual point about the relationship between SAE architecture and assumptions about the data is arguably more important.**
>
> Thank you for raising this important point! As rightly pointed out by the reviewer, the primary goal of our paper is to highlight the biases in SAEs that arise due to their architecture, limiting the concepts they can recover from data. We analytically obtain the assumptions of popular SAEs, and propose SpaDE as a concrete example of constructing an SAE incorporating specific properties of the data—nonlinear separability and concept heterogeneity—to highlight the limitations of existing SAEs. We perform extensive empirical analyses across multiple modalities to validate our claims: by training existing SAEs and SpaDE on synthetic gaussian clusters data, semi-synthetic nanoGPT activations on formal languages, and naturalistic DINOv2 model activations on ImageNette. Our experimental analysis includes sparsity-reconstruction frontiers (Figures E2, E7, E13 in Appendix). We plan to scale up our experiments with SpaDE to full-scale LLMs in future work.
>
> ---
> ---
>
> > **Do you have any thoughts on the applicability of your framework to more general types of interpreter models?**
>
>
> We share your intuition! Our main contribution in this paper is that SAEs are biased toward identifying concepts that match their prior assumptions. Our insights into SAE biases are a property of the encoder architecture, specifically of the pre-image of the encoder map, which is formalized using receptive fields of encoder latents (Definition 3.2, line 110). The insights into biases of encoder architecture will thus directly translate to Cross-Layer Transcoders (CLTs) that use the same encoder, but are trained on a different objective. For instance, a ReLU-CLT or JumpReLU-CLT both make the same assumption we highlight for (Jump)ReLU-SAEs: concepts are linearly separable from the rest of the data in the input layer to the CLT. This insight about the interpreter architecture limiting the concepts that can be recovered is also true in greater generality and has to do with the expressive power of the tool—any tool that seeks to interpret model latents by decomposing them into monosemantic, human-understandable concepts, will be able to do so only for certain kinds of concepts that its architecture allows. Being aware of this allows us to think carefully about the structure of concepts in concept space and design tools that are expressive enough to recover concepts of interest.
>
>
>
> > **Do you think your work could explain any potentially-pathological results that have been described about SAEs? Could our choice of architecture be leading to feature splitting, or to non-atomic features being learned, that could be ameliorated with a different choice of architecture?**
>
> Great point! Through our work, we seek to highlight the assumptions made by SAEs about the structure of concepts. While we do not discuss this in our paper, it is likely that pathological results using SAEs, such as feature splitting, may be a consequence of a mismatch between the concept structure in data (model activations) and the structure assumed by specific SAEs. There may also be other contributing factors to feature splitting such as choice of SAE width, training objective (sparsity regularizer), etc.
>
> **Summary**: We thank the reviewer for their review and suggestions! We would be happy to answer any further questions during the discussion period.

---

> > ### Comment · Reviewer_E5bk · 2025-08-02
> >
> > Thanks for the response! I'll keep my rating at 5. I think the paper is interesting and well-executed.

---

### Official Review · Reviewer_nypq · 2025-07-03

**Clarity:** 3
**Significance:** 3
**Originality:** 2
**Rating:** 5
**Confidence:** 3

**Summary:**

The authors challenge the common assumption that Sparse Autoencoders (SAEs) are neutral tools for uncovering meaningful concepts in neural representations. By recasting SAE training as a bilevel optimisation problem, they show that each SAE architecture implicitly projects data onto a specific geometric constraint set - e.g., the positive orthant (ReLU), sparse angular subspaces (TopK), or hypercubes (JumpReLU). These constraints shape the encoder's receptive fields, biasing the model towards detection of concepts that conform to assumptions, such as linear separability and fixed intrinsic dimensionality. The authors conduct a series of experiments across a range of settings to empirically verify this framework. They propose a new architecture SpaDE, which uses the sparsemax nonlinearity to handle nonlinear separability and concept heterogeneity, and demonstrate it consistently outperforms other SAEs in recovering monosemantic, interpretable features.

**Questions:**

Co-occurence and overlapping concepts: As noted by the authors in Section 6, the analysis focuses on mutually exclusive concepts, where the presence of one concept implies the absence of others. While the central argument about SAEs being biased towards certain concepts due to their architectural constraints clearly holds in general, regardless of the data distribution, I'm curious how the empirical failure modes (e.g., lack of monosemanticity) differ qualitatively in these non-exclusive scenarios (e.g., where concepts are overlapping/compositional)? In particular, do real-world concept distributions tend to violate the separability and dimensionality assumptions identified? I would expect this to be the case, but it would be valuable to understand whether these failure trends persist in these non mutually exclusive settings.

**Ethical Concerns:**

["NO or VERY MINOR ethics concerns only"]

**Final Justification:**

The rebuttal clarified SpaDE’s intended scope, addressed generalisation to overlapping concepts, and acknowledged its own geometric assumptions. With no substantive issues remaining, I maintain my positive assessment and recommendation for acceptance.

**Limitations:**

Yes, very well elaborated on in Section 6.

**Quality:**

3

**Strengths And Weaknesses:**

**Strengths:**
- Focus on latent space geometry and implications for interpretability research: connecting the encoder receptive field to assumptions about the underlying geometry of the concepts provides a compelling argument that latent space geometry must be prioritised when designing SAEs. The paper makes it clear that this is a fundamental limitation of current SAEs, and the theoretical framing is timely and well-needed in the interpretability field.
- Formalisation via bi-level optimisation: the paper presents a principled reformulation of SAE training as a bilevel optimisation problem, where the encoder implicitly solves a projection problem over an architecture-specific constraint set. This offers a clean, mathematically grounded explanation for how architectural choices induce biases in feature extraction, and is broadly applicable to multiple existing SAE variants.
- Broad range of evaluation settings: experimental evaluations spans a wide range of settings - from synthetic separability setups to semi-synthetic formal language models and real-world visual representations. The broadness of the evaluation approach provides strong support for the generality of the claims.
- Principled design of SpaDE: the architecture proposed is not just proposed heuristically, but instead motivated by the geometric limitations identified in the earlier framework. SpaDE is also shown to resolve failure modes observed in prior models.
- Clarity of empirical failures: the empirical results convincingly demonstrate how specific structural assumptions (e.g., linear separability, fixed concept dimensionality) lead to systematic failure modes in prior SAEs, especially in more complex settings.

**Weaknesses:**
- Assumptions underlying SpaDE not extensively tested: as highlighted by the authors in Section 6, SpaDE assumes that Euclidean distance is a meaningful distance metric in concept space, and that concepts project well onto the probability simplex. These assumptions may not hold across domains with more complex geometries, however, the authors do acknowledge this and this is likely to be more diagnostic of a broader, open problem, as much of the current literature implicitly assumes Euclidean geometries.

---

> ### Author Rebuttal · Authors · 2025-07-30
>
> We thank the reviewer for their review and comments! We love the reviewer’s framing of our main message: “a compelling argument that latent space geometry must be prioritised when designing SAEs”. We appreciate the description of our work as a “principled reformulation of SAE”, “timely and well-needed” having a “broad range of evaluation settings”. We also appreciate that the reviewer highlighted our main contributions: 1) SAE architecture constraints bias the model towards concepts that conform to its assumptions, and 2) handling nonlinear separability and concept heterogeneity using SpaDE, a geometry-driven SAE.
>
> We provide answers to specific questions/comments below.
>
>
> > **Assumptions underlying SpaDE not extensively tested: as highlighted by the authors in Section 6, SpaDE assumes that Euclidean distance is a meaningful distance metric in concept space, and that concepts project well onto the probability simplex. These assumptions may not hold across domains with more complex geometries, however, the authors do acknowledge this and this is likely to be more diagnostic of a broader, open problem, as much of the current literature implicitly assumes Euclidean geometries.**
>
>
> This is a great point! As noted by the reviewer, our main goal in this work was to highlight the assumptions made by existing SAEs about concept geometry, and the inability of SAEs to recover concepts when their assumptions do not match the structure of concepts in model activations. To this end, we present SpaDE as a concrete example of designing an SAE which can capture two reasonable data properties—nonlinear separability and concept heterogeneity—that other SAEs fail to capture. Using our insights on SpaDE indicates that SpaDE also makes assumptions about concept structure: SpaDE assumes that Euclidean distance is useful in concept space (as rightly pointed out by the reviewer). Since we use SpaDE as a useful baseline to highlight the limitations of existing SAEs, as the reviewer noted, we *do not* claim that SpaDE is a superior SAE. We merely aim to suggest that when the data follows priors assumed by SpaDE (or for that matter any other SAE), it can be expected to perform better.
>
> ---
> ---
>
> > **While the central argument about SAEs being biased towards certain concepts due to their architectural constraints clearly holds in general, regardless of the data distribution, I'm curious how the empirical failure modes (e.g., lack of monosemanticity) differ qualitatively in these non-exclusive scenarios (e.g., where concepts are overlapping/compositional)? In particular, do real-world concept distributions tend to violate the separability and dimensionality assumptions identified?**
>
> Thank you for this observation! As rightly highlighted by the reviewer, our claim about SAEs being biased towards certain concepts is a property of their architecture, and thus holds for any data distribution. We use non-overlapping concepts as a simple case with real-world examples (such as mutually exclusive classes or children of the same parent in a hierarchical generative process) to highlight the limitations of existing SAEs across diverse domains—from synthetic gaussian clusters to semi-synthetic nanoGPT activations on formal languages to naturalistic DINOv2 model activations from ImageNette. Our claims about SAE biases will also hold for overlapping concepts if we generalize the definition of separability from (current) *distinguishing one concept from other concepts (akin to multiclass classification)* to (improved) *distinguishing between the presence and absence of a concept (binary classification for each concept)*.
> We will formally highlight the significance of SAE biases in the presence of overlapping concepts in future work. As for the real-world concept distributions that violate separability and dimensionality assumptions, we highlight examples from the literature on lines 133, 139 and 140.
>
> **Summary**: We thank the reviewer again for their thoughtful comments! We would be happy to answer any further questions during the discussion period.

---

> ### Comment · Reviewer_nypq · 2025-08-03
>
> Thank you to the authors for their very thoughtful rebuttal. I appreciate the clarification that SpaDE is not intended as a universally superior SAE, but rather as a geometry-motivated ex. illustrating how nonlinear separability and concept heterogeneity (two properties that challenge other SAEs) can be addressed within a principled design framework.
>
> My original questions were largely aimed at high-level considerations, particularly how the framework might extend to overlapping or compositional concepts, and how assumptions about geometry play out across different domains, rather than concerns about the technical soundness of the work. The authors’ discussion here is helpful: reframing separability as distinguishing presence vs. absence of a concept is a natural and promising extension, and I look forward to seeing it developed more formally in future work. I also value the explicit acknowledgment that SpaDE itself makes non-universal assumptions (e.g., Euclidean distance, projection onto the simplex).
>
> Overall, the rebuttal addresses my main points of uncertainty. My assessment of the paper remains very positive: the work is technically solid, makes a timely and principled contribution to interpretability research, and raises an important discussion about the role of latent space geometry in SAE design. I maintain my original score and recommendation for acceptance.

---

### Official Review · Reviewer_9gLg · 2025-07-03

**Clarity:** 3
**Significance:** 4
**Originality:** 4
**Rating:** 5
**Confidence:** 3

**Summary:**

The authors investigate the limitations of Sparse Autoencoders (SAEs) in terms of the types of concepts they can detect. They attribute these limitations to the architectural design of SAEs, which imposes specific assumptions on the structure how features are encoded. To address these constraints, the authors propose a new SAE, SpaDE, which is able to address previous limitations on data assumptions.

**Questions:**

- How do you interpret the occurrence of more different colored clusters in SpaDE compared to the others?

- When analyzing SAEs, which true structure of the data should be considered?

- Does good data selection for analysis always depend on the task?

**Ethical Concerns:**

["NO or VERY MINOR ethics concerns only"]

**Final Justification:**

I recommend accepting the paper. The topic is timely and relevant, and the paper is well developed.

**Limitations:**

Yes.

**Paper Formatting Concerns:**

For Figures 1-3, 5-6, and all Tables 1-2, captions should be lower case (except for first word and proper nouns).

**Quality:**

4

**Strengths And Weaknesses:**

# Strengths

- The paper is well-written, logically structured, and easy to follow. Each step is clearly motivated and illustrated.

- The work is timely and addresses a relevant problem, offering a better understanding of shortcomings of SAEs.

- The figures are overall clear, well-designed, and effectively support the main points.

# Weaknesses

- The domain of the work is not clearly specified in the first part of the paper and only becomes explicit in Sections 5.3 and 5.4. Referencing this earlier would improve clarity.

- It is not entirely clear what is meant with "true structure" of concepts (l. 126).

- In Figure 5, row (c), the meaning of the different cluster colors is unclear and should be clarified in the caption or main text.

---

> ### Author Rebuttal · Authors · 2025-07-30
>
> We thank the reviewer for their review and feedback! We appreciate the reviewer’s assessment of our work as “timely”, “addresses a relevant problem” and our paper as “well written”, “easy to follow” with “well-designed” figures. We thank the reviewer for highlighting our paper’s main contributions: 1) limitations of SAEs arise from architecture of SAEs imposing assumptions on concept structure, 2) highlighting how specific limitations can be addressed through SpaDE.
>
> We provide responses to specific comments/ questions by the reviewer below.
>
> > **The domain of the work is not clearly specified in the first part of the paper and only becomes explicit in Sections 5.3 and 5.4. Referencing this earlier would improve clarity.**
>
> Thank you for the suggestion! We emphasize that since our theoretical claims did not require assumptions specific to a given modality, we intentionally framed our paper to be domain-agnostic in its narrative. This is because our main contribution—the duality between geometric structure of concepts in model activations and an SAE that recovers these concepts—is general and not constrained to specific domains. This is demonstrated in our experiments that cover a broad range of experimental settings, ranging from synthetic datasets (gaussian clusters, Sections 5.1, 5.2), to semi-synthetic datasets (nanoGPT activations on formal languages, Section 5.3), and to naturalistic settings (DINOv2 on ImageNette, Section 5.4).
>
> Since Sparse Autoencoders (SAEs) are being used across multiple domains for interpreting neural networks, including language models (examples in line 30), vision models (line 31), and protein autoregressive models (line 32), we believe our claims can be meaningful and relevant for all such communities. We hence believe having a domain-agnostic narrative is ideal for our work.
>
> > **It is not entirely clear what is meant with "true structure" of concepts (l. 126).**
>
> Fair point. By the phrase “true structure” of concepts, we intended to make an explicit distinction between the ground truth structure in accordance with which concepts are organized in a model’s activations versus the structure assumed by different SAEs. By making this distinction, we wish to highlight a core contribution of our work: that SAEs make assumptions about concept structure that may not hold in the ground truth structure in model activations, and an SAE cannot recover concepts whose “true structure” in model activations does not match the “assumed structure” of concepts by that SAE. We will make sure to clarify the writing to better emphasize what we mean by “true structure” in the paper.
>
>
> > **In Figure 5, row (c), the meaning of the different cluster colors is unclear and should be clarified in the caption or main text.**
>
> Thank you for pointing this out! Figure 5 demonstrates the limitations of SAEs occurring due to a mismatch between SAE assumptions and ground-truth concept structure, when trained on data with nonlinearly separable concepts constructed using synthetic gaussian clusters.  While rows (a) and (b) quantitatively and qualitatively show the monosemanticity (concept selectivity) of different latents in each SAE, row (c) captures the monosemanticity of the entire sparse code of all SAEs.
> The matrix in row (c) shows the cosine similarity between sparse codes of pairs of datapoints—ideally, different latents activate for different concepts leading to block-diagonal structure in this matrix. The cluster colors in row (c) are the results of spectral clustering on this matrix of pairwise cosine similarities: all points with the same cluster color on this plot activate a common set of SAE latents. Having multiple concepts (gaussian clusters) with the same color indicates that the SAE is unable to distinguish these concepts well in its sparse code, and thereby unable to monosemantically recover these concepts. As predicted by our analysis, ReLU and JumpReLU SAEs show multiple clusters with the same color, and are hence unable to recover nonlinearly separable clusters. We will include a brief intuitive description of the meaning of cluster colors in the final version of the paper.
>
> ---
> ---
>
> > **How do you interpret the occurrence of more different colored clusters in SpaDE compared to the others?**
>
> Great question, we assume this question is in reference to the different colored clusters in SpaDE in Figure 5c (please correct if we are misinterpreting the comment)!
> Specifically, Figure 5 is aimed at highlighting the limitations of SAEs due to a mismatch between SAE assumptions (linear separability assumed by ReLU and JumpReLU SAEs) and the property of nonlinear separability. Accordingly, points which share the same color in Figure 5 (c) are those that activate a common set of SAE latents. The occurrence of different colors for different clusters (concepts) in SpaDE, while ReLU and JumpReLU SAEs assign similar colors to different clusters, highlights the inability of ReLU and JumpReLU SAEs in recovering nonlinearly separable concepts in a monosemantic fashion.
> Having multiple concepts share the same color indicates that SAE latents are polysemantic (respond to multiple concepts), which is clearly observed for ReLU and JumpReLU SAEs (and TopK SAE to some extent). SpaDE shows different concepts (gaussian clusters) with different colors, indicating that the latents learnt by SpaDE are more monosemantic than the other SAEs when concepts are nonlinearly separable. It is worth noting that SpaDE is meant to be a baseline to show that the limitations arising in existing SAEs are specific to their architecture, and can be avoided.
>
>
> > **When analyzing SAEs, which true structure of the data should be considered?**
>
> Thank you, we believe this is exactly the right question to ask after reading the paper, and it’s one that has sparked extensive debate among the authors!
>
> To answer directly: there may be no free lunch here. The true structure of model activations is not fixed, it depends on the architecture, training regime, and task. As such, we believe no single SAE architecture is likely to perform well across all settings. Instead, the question of which structure the data contains should guide the choice of SAE.
> This is the central message of our work: there exists a duality between the geometry of the data and the inductive biases of the SAE used to probe it. Some SAEs assume linear separability, others enforce fixed-dimensional sparsity, and each of these choices determines not just what can be extracted, but what is even visible.
>
> A corollary to this is that we should therefore begin by asking: what is the likely structure of this representation space? Only then does the choice of SAE become meaningful. And more broadly, **we hope this work encourages the field to invest more deeply in understanding the geometric organization of latent spaces**. Much effort has gone into designing SAEs, but comparatively little has been spent understanding the structure they operate on.
>
>
> > **Does good data selection for analysis always depend on the task?**
>
> Thank you for the question. Unfortunately, we are slightly unclear about how to interpret it, but provide a response based on the following (paraphrased) interpretation: why did we choose the specific synthetic modalities included in the paper (which are gaussian clusters) to analyse SAEs?
>
> We note that our main goal in the experiments was to highlight the biases/limitations of existing SAEs, for which we designed synthetic data to capture specific properties–nonlinear separability and concept heterogeneity–that are complementary to the assumptions of existing SAEs. Our choice of datasets was motivated to study the SAEs better and bring out their shortcomings. The SAEs can themselves be used in any task (interpreting language models/ vision models/ protein autoregressive models, etc) and the limitations we highlight will persist.
>
> If this interpretation is different from what the reviewer had in mind, we would be happy to hear back from them and provide an updated response!
>
>
> > **For Figures 1-3, 5-6, and all Tables 1-2, captions should be lower case (except for first word and proper nouns).**
>
> Thank you for pointing this out! We will make the changes in the final version of the paper.
>
> **Summary**: We thank the reviewer again for their questions and suggestions! We would be happy to address any further questions during the discussion period.

---

> > ### Comment · Reviewer_9gLg · 2025-08-02
> >
> > Thank you for addressing my questions. I believe the paper is solid, and I will maintain my score of 5.

---

### Note · Authors · 2025-08-11

Dear Reviewers,

We thank you all for your thorough reviews, interesting insights and constructive criticism of our paper! We are pleased that the reviewers found our work to be “solid”, "timely", “addresses a relevant problem” (R. 9gLg), “a compelling argument that latent space geometry must be prioritised while designing SAEs”, having a “broad range of evaluation settings” (R. nypq), “core point … is highly valuable” (R. E5bk), and “valuable theoretical perspective” (R. Bs68). We also appreciate that the reviewers liked the presentation of the paper, calling it “well-written”, “easy to follow” (R. 9gLg), “with beautiful figures” (R. E5bk), and “presentation quality is excellent” (R. Bs68).

To accommodate reviewers’ feedback, particularly about the following points:
- difference between compositions of atomic (1-D) concepts and inherently multidimensional concepts (R. Bs68);
- real world applicability of SpaDE and associated limitations (R. Bs68);
- further clarifying assumptions underlying SpaDE (R. nypq);
- description of figures (esp. Fig 5) (R. 9gLg);
- formatting suggestions and minor comments (R. 9gLg, R. Bs68); and
- intuition for theoretical claims (R. Bs68),

we plan to use the extra page in the camera-ready version to include further discussion and clarification in Sections 5 and 6, as well as other appropriate sections, of the paper.
Finally, we thank the reviewers, as well as the ACs and everybody involved in the review process, for their valuable time invested in reviewing our paper!

---

### Decision · Program_Chairs · 2025-09-17

**Decision:**

Accept (poster)

**Comment:**

This paper analyzes how Sparse Autoencoders (SAEs) are constrained by architectural assumptions, framing training as a bilevel optimization problem and introducing **SpaDE**, which addresses nonlinear separability and heterogeneity. The work is timely, clearly presented, and supported by strong figures and broad experiments spanning synthetic clusters, semi-synthetic formal languages, and naturalistic vision settings.

### Strengths
- Principled theoretical framing.
- Clear link between geometry and concept recovery.
- Broad range of evaluations across modalities.
- SpaDE as a concrete, geometry-motivated example design.

### Limitations
- SpaDE itself makes assumptions (Euclidean geometry, simplex projection).
- Large-scale LLM experiments and interpretability benchmarks are lighter.
- Distinction between atomic vs. multi-dimensional concepts could be clearer.
- Practical significance remains somewhat open.

### Rebuttal
The authors clarified that SpaDE is not positioned as a universal SAE but as a baseline illustrating the duality between concept geometry and SAE design. They acknowledged SpaDE’s assumptions explicitly, reframed separability for overlapping concepts, and explained why non-linear concepts must sometimes be directly recovered rather than decomposed. These clarifications addressed most reviewer concerns and reinforced the paper’s conceptual contribution.

### Decision
Three reviewers gave **strong accept (score 5)** and one gave a **borderline accept (score 4)**. No reviewer recommended rejection, and the rebuttal further strengthened confidence in the contribution. The consensus is clearly positive, with reviewers consistently praising the clarity, conceptual framing, and breadth of evaluation.

**Recommendation: Accept.** This is a principled and high-quality contribution that deepens our understanding of SAE limitations and will likely influence future interpretability research.